# DATA PRUNING BY INFORMATION MAXIMIZATION

**Haoru Tan** [*][1], **Sitong Wu**[*][2], **Wei Huang**[1], **Shizhen Zhao**[1], **Xiaojuan Qi**[†][1]

[1]The University of Hong Kong       [2]The Chinese University of Hong Kong

## ABSTRACT

In this paper, we present InfoMax, a novel data pruning method, also known as coreset selection, designed to maximize the information content of selected samples while minimizing redundancy. By doing so, InfoMax enhances the overall informativeness of the coreset. The information of individual samples is measured by importance scores, which capture their influence or difficulty in model learning. To quantify redundancy, we use pairwise sample similarities, based on the premise that similar samples contribute similarly to the learning process. We formalize the coreset selection problem as a discrete quadratic programming (DQP) task, with the objective of maximizing the total information content, represented as the sum of individual sample contributions minus the redundancies introduced by similar samples within the coreset. To ensure practical scalability, we introduce an efficient gradient-based solver, complemented by sparsification techniques applied to the similarity matrix and dataset partitioning strategies. This enables InfoMax to seamlessly scale to datasets with millions of samples. Extensive experiments demonstrate the superior performance of InfoMax in various data pruning tasks, including image classification, vision-language pre-training, and instruction tuning for large language models.

## 1 INTRODUCTION

Large-scale datasets have been pivotal in the recent breakthroughs in deep learning (Brown, 2020; Kirillov et al., 2023; Radford et al., 2021; Rombach et al., 2022). However, the growing size of training data substantially increases both training costs and storage demands. Moreover, significant redundancies within these datasets highlight the importance of data-pruning methods that remove redundant samples and identify a compact yet informative subset, known as a coreset, to enable more efficient model training and data storage.

Research in this field can be broadly divided into two categories: score-based methods (Sorscher et al., 2022; Radford et al., 2021) and geometry-based methods (Sener & Savarese, 2017; Ash et al., 2020). Score-based methods focus on developing metrics to evaluate a sample's informativeness, such as prediction uncertainty (Har-Peled et al., 2007), loss value (Cody Coleman et al., 2019), or influence score (Xia et al., 2024; Tan et al., 2023). However, as shown in Figure 2(a), these methods often select samples densely concentrated in regions with the highest scores, leading to redundancies and failing to consider simpler samples with lower scores, which results in biased selections. Recent work (Sorscher et al., 2022) highlights that even simple samples are important for improving model generalization. On the other hand,

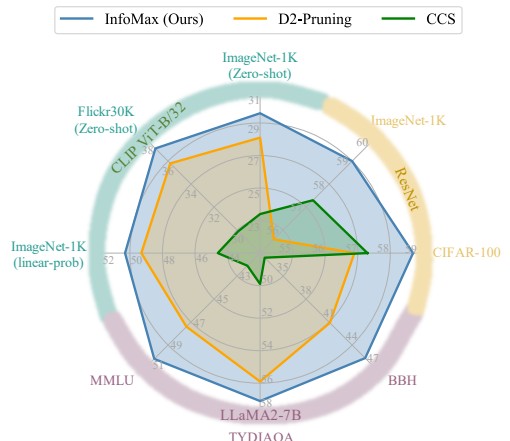

Figure 1: Summarization of InfoMax's performance in vision-language pre-training and image classification (both at 10% selection ratio), and instruction fine-tuning for large language models (5% selection ratio). The results show that InfoMax has demonstrated substantial progress in various scenarios, see Section 4 for details.

---

*Equal Contribution. [†] Corresponding Author

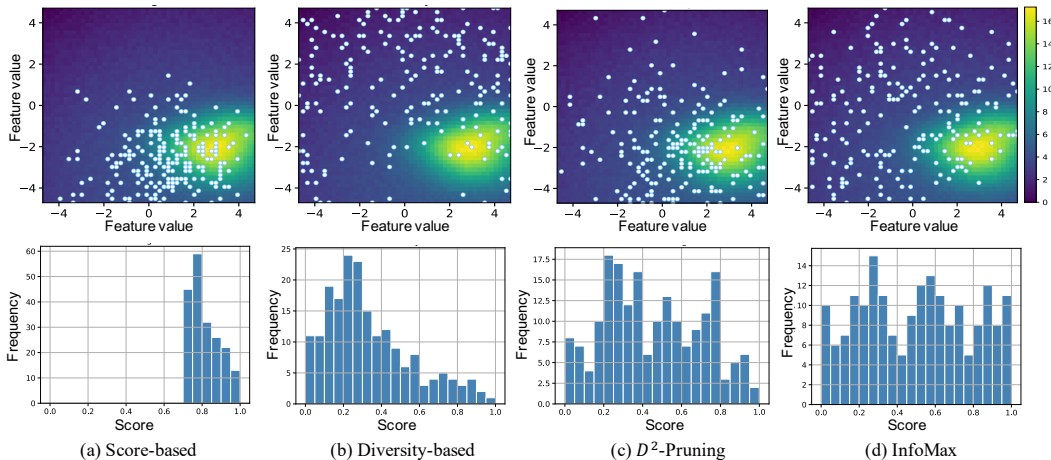

(a) Score-based     (b) Diversity-based     (c) $D^2$-Pruning     (d) InfoMax

Figure 2: The coreset distributions from InfoMax, and, the score-based method (output margin (Har-Peled et al., 2007), and diversity-based method (k-Median clustering (Feldman & Langberg, 2011; Har-Peled & Mazumdar, 2004)), hybrid method ($D^2$-Pruning (Maharana et al., 2023)). In the upper figures, the background illustrates the distribution density of the data in the space after PCA dimensionality reduction. Brighter colors indicate a higher density of samples. Simultaneously, we present the locations of the samples selected by different methods using scatters. The figures below show the distribution of the scores of the corresponding coreset. This set of experiments is conducted on CIFAR-100 (Krizhevsky, 2009). The coreset selected by our InfoMax is capable of covering both high-density and low-density regions in terms of spatial uniformity, and also high-information and low-information data in terms of score distribution.

geometry-based methods aim to select a diverse subset of the data, reducing redundancies between samples (Sener & Savarese, 2017). As illustrated in Figure 2(b), while these methods prioritize diversity, they often overlook informative samples with high-importance scores, leaving a large number of low-scoring samples in the selection.

Recently, hybrid approaches have emerged that combine importance scores with diversity to design more effective algorithms (Ash et al., 2020; Maharana et al., 2023; Zheng et al., 2022; Yang et al., 2024). One of the most notable examples is $D^2$-Pruning (Maharana et al., 2023), which models a dataset as a graph. In this framework, node scores represent a sample's informativeness, such as difficulty or importance, while edges capture the similarities between samples. The data pruning process is formulated as an iterative node selection problem, where at each step, nodes with the highest scores are selected, and the scores of neighboring nodes are reduced to account for redundancy. However, due to its greedy selection process, the algorithm is prone to getting stuck in suboptimal solutions, making it challenging to maintain a proper balance between importance and diversity. For example, as shown in Figure 2(c), many samples remain concentrated in high-density areas with low scores, leaving significant portions of the space uncovered.

In this paper, we introduce *InfoMax*, a new and effective approach for data pruning and coreset selection. Our core insight is to find a subset of samples that maximizes overall information by simultaneously considering each sample's information contribution and the information overlap among them. First, a sample's informativeness can be explained by its importance or difficulty. For instance, a sample with intricate structures, complex backgrounds, and occlusions provides more information than one with simple patterns see Figure 3. Therefore, we explore score-based metrics that evaluate a sample's difficulty or importance, providing an information assessment. Second, the information overlap among samples is evaluated in a pairwise manner and quantified by their pairwise similarities. Samples with greater

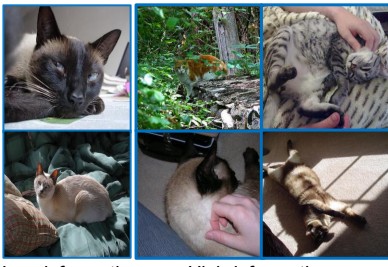

Low-informative      High-informative

Figure 3: Cases of samples with low informativeness (left) and high informativeness (right).

similarity will have a higher degree of information overlap. Using these considerations, we formulate coreset selection as a discrete quadratic programming (DQP) problem with equality constraints that

specify the desired number of selected samples, see Eq. (2). The objective function represents the overall information as the sum of each sample's individual information, reduced by redundancies introduced by the presence of similar samples within the coreset. Furthermore, we propose an efficient and robust gradient-based solver to address scalability. This solver is enhanced by sparsification techniques applied to the similarity matrix and dataset partitioning strategies, enabling practical and scalable computation. For instance, our method processes 12 million data points in just 37 minutes (see Section B.2 for details). By solving this problem, InfoMax identifies a subset of samples that maximizes overall information, thereby reducing the likelihood of suboptimal solutions often seen with greedy-based methods, see the discussion in Section 3.3. As shown in Figure 2(b), the coreset generated by our approach strikes a good balance between diversity (i.e., well-distributed across the entire space) and informativeness (i.e., containing a reasonable number of important samples). The superior performance across multiple tasks, including image classification, vision-language pretraining, and instruction fine-tuning for large language models, see Section 4, further supports the effectiveness of our new formulation. We have summarized the overall pipeline in Figure 4 and Algorithm 1. In summary, our contributions are:

- We propose InfoMax, a new coreset algorithm designed to maximize overall information by accounting for each sample's individual contribution while reducing information overlap, with a simultaneous focus on maintaining diversity and importance.

- We propose an efficient gradient-based solver enhanced by sparsification techniques and dataset partitioning strategies to make InfoMax scale to large-scale datasets.

- Extensive experiments show that InfoMax exhibits the best performance and consistently outperforms the state-of-the-art schemes in a series of tasks, including image classification, and vision-language pre-training (Radford et al., 2021), large language model supervised fine-tuning (Xia et al., 2024) experiments. Notably, it brings about a significant improvement of approximately 5.5% compared to the previous best methods at a 5% selection rate in instruction fine-tuning experiments. Additionally, it shows around 2% performance enhancements on classification tasks at a 10% selection rate, see Figure 1.

## 2 PRELIMINARIES

In this section, we first present the problem definition for data pruning, followed by a discussion of existing methods, including score-based, diversity-based, and hybrid approaches.

### 2.1 PROBLEM DEFINITION

Before diving into the literature review of existing methods, we first define the problem of data pruning, also known as coreset selection (Sener & Savarese, 2017; Mirzasoleiman et al., 2020; Killamsetty et al., 2021). Let $D = (z_i)_{i=1}^{N}$ represent the training set, drawn independently and identically distributed (i.i.d.) from an underlying data distribution. The goal of dataset pruning is to identify the most informative subset of the training data that minimizes information loss while maximizing model performance. Formally, the problem can be stated as:

$$\mathbf{S}^* = \arg \max_{\mathbf{S} \subset \mathbf{D}, |\mathbf{S}|=p} I(\mathbf{S}), \tag{1}$$

where $p$ is the budget of the target coreset and $|\cdot|$ represents the cardinality of a set. $I(\mathbf{S})$ measures the set-level information of the candidate subset $S$. There are multiple choices (Sorscher et al., 2022; Wei et al., 2015; Kaushal et al., 2021) for the instantiation for the set information $I(\mathbf{S})$. For example, given the loss function $l(\cdot)$ and the network parameters $\mathbf{w}$, previous works Yang et al. (2023); Zheng et al. (2022) often use the overall test loss reduction caused by training the model on $S$ as the information measure $I(\mathbf{S}) = \mathbb{E}_{z \sim P}\left[l(z, \mathbf{w}^0) - l(z, \mathbf{w}_{\mathbf{S}}^*)\right]$, Eq. (1) would be identical as $\mathbf{S}^* = \arg \min_{\mathbf{S} \subset D, |\mathbf{S}|=p} \mathbb{E}_{z \sim P}\left[l(z, \mathbf{w}_{\mathbf{S}}^*)\right]$, where $\mathbf{w}^0$ is the initial parameter and $\mathbf{w}_{\mathbf{S}}^*$ indicates the optimal network parameter learned on $\mathbf{S}$. The key distinction between various data pruning methods lies in how they define $I(\mathbf{S})$, which is detailed below.

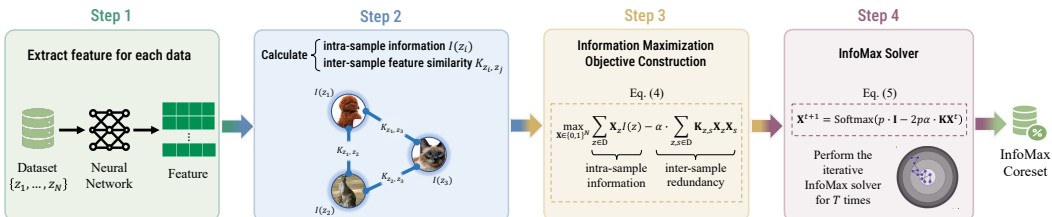

Figure 4: The overall pipeline of our InfoMax. In the first two steps, we use a network to extract features to calculate the similarity matrix and the intra-sample information terms. Then, in step 3, we construct the quadratic optimization problem for infoMax, see Eq. (2) for details. Finally, we perform the iterative InfoMax solver as defined by Eq. (3) to obtain the final InfoMax coreset.

## 2.2 Score-based Method

The score-based method (Paul et al., 2018; Tan et al., 2023; Toneva et al., 2018) often selects samples solely based on the score values. They generally model $I(\mathbf{S})$ as: $I(\mathbf{S}) = I(z_1) + I(z_2) + ... + I(z_p)$, where $\mathbf{S} = \{z_1, \ldots, z_p\} \subset \mathbf{D}$, with each $z$ representing a data sample, and $I(z)$ denoting the information associated with that sample. The primary task then becomes identifying a suitable score metric to evaluate a sample's individual contribution, $I(z)$, of each sample. Various scores have been proposed to quantify $I(z)$, such as model uncertainty (Har-Peled et al., 2007), loss value (Cody Coleman et al., 2019), and influence score (Tan et al., 2023; Pruthi et al., 2020).

The assumption underlying the score-based formulation is that there is no information overlap between different samples, allowing the individual contributions of samples to be summed to represent the overall contribution. However, this approach fails to account for overlap in the information provided by different samples. For instance, if two identical samples, $z_i$ and $z_j$, both receive high scores, the information gained from adding $z_j$ after selecting $z_i$ would be negligible. As a result, this method cannot ensure that the selected samples offer broad coverage of the data space, leading to a loss of diversity and suboptimal solutions. This limitation is illustrated in Figure 2(a), they select samples densely concentrated in regions with the highest scores, leading to redundancies and failing to consider simpler samples with lower scores, which results in biased selections.

## 2.3 Geometry-based Method and Hybrid Method

**Geometry-based Method** In contrast to score-based methods, geometry-based methods design $I(\mathbf{S})$ to consider maximizing the diversity and minimizing the information overlap among samples. For many works (Lloyd, 1982; Tan et al., 2006; Coates & Ng, 2012; Har-Peled & Mazumdar, 2004; Feldman & Langberg, 2011), the set-level information $I(\mathbf{S})$ is regarded as: $I(\mathbf{S}) \propto \sum_{(z_i, z_j) \in \mathbf{S}} -I(z_i; z_j)$, where $I(z_i, z_j)$ measures the similarity of two samples, indicating information overlap (also the mutual information). To solve the above problem, Sener & Savarese (2017) applied greedy k-center to choose the coreset with good data coverage.

This formulation assumes that all individual samples carry equal amounts of information, disregarding the varying significance of each sample. Consequently, it tends to overlook critical samples while retaining a large number of non-informative ones. As illustrated in Figure 2(b), while these methods prioritize diversity, they often overlook informative samples with high-importance scores, leaving a large number of low-scoring samples in the selection.

**Hybrid Method** More recently, hybrid methods (Zheng et al., 2022; Ash et al., 2020; Maharana et al., 2023) have been developed that account for both the individual importance and diversity of samples simultaneously. For instance, CCS (Zheng et al., 2022) sets different sampling ratios for samples with different scores to enhance data coverage and balance both easy and hard samples. $D^2$-Pruning (Maharana et al., 2023) proposes to select data on the graph. One of its core steps is the Inverse Message Passing operation. Specifically, it selects the sample with the highest score among all unselected candidates and subsequently reduces the scores of the neighborhood. However, the above approaches are primarily based on heuristic designs and are often solved using greedy algorithms. These methods are prone to local optima, as the algorithm lacks a holistic view of the problem at each step, resulting in suboptimal outcomes. In Figure 2, the selected samples demonstrate

inadequate coverage of the entire space, with many concentrated in low-importance regions, hence results in the suboptimal performance in practice, see Figure 1 for details.

In this paper, we propose a unified formulation with a scalable solver for coreset selection, aimed at maximizing the information of the selected subset by accounting for both the individual contributions of samples and their redundancies.

# 3 METHOD

## 3.1 INFOMAX: FORMULATIONS AND SOLUTIONS

The objective of InfoMax is to identify a subset of data samples that maximize intra-sample information while minimizing inter-sample redundancies caused by similar samples, thereby achieving an optimal balance between diversity and importance. For a sample $z$, we represent its information content as $I(z)$. For two samples $z$ and $s$, their information overlap or redundancy is denoted as $I(z, s)$. The total information is measured by summing the individual information of the samples, with deductions for inter-sample redundancy, as outlined below. We will discuss in Section 3.3 that under some mild settings, this optimization problem in Eq. (2) is equivalent to solving the original information maximization problem for data pruning as defined in Eq. (1).

**Problem Formulation for InfoMax** Here, we consider inter-sample redundancy in a pairwise manner using $I(z, s)$ and the matrix containing all pairwise sample similarities is denoted as $\mathbf{K}_{z,s}$. Then, given the dataset $\mathbf{D} = \{z_1, z_2, ..., z_N\}$, we introduce a variable $\mathbf{X}_z \in \{0, 1\}$ to represent whether a sample is selected ($\mathbf{X}_z = 1$) or pruned ($\mathbf{X}_z = 0$), the overall information is formulated as a quadratic function of variable $X$ as:

$$\max_{\mathbf{X} \in \{0,1\}^N} \sum_{z \in \mathbf{D}} \mathbf{X}_z I(z) - \alpha \cdot \sum_{z,s \in \mathbf{D}} \mathbf{K}_{z,s} \mathbf{X}_z \mathbf{X}_s, \quad \text{s.t.} \quad \sum_{z \in \mathbf{D}} \mathbf{X}_z = p, \tag{2}$$

where $p = (1 - \delta)N$ is the size budget of the selected set. In the formulation, the first-order term $I(z)$ measures the intra-sample information of the sample $z$, which can be instantiated using any existing sample-wise scores, e.g., EL2N (Paul et al., 2018) or SSP (Sorscher et al., 2022). The quadratic term $\mathbf{K}_{z,s}$ measures the inter-sample redundancy between the sample $z$ and $s$.

**Objective Construction** For the intra-sample information $\mathbf{I} \in \mathbb{R}^N$ and the pairwise feature similarity $\mathbf{K} \in \mathbb{R}^{N \times N}$, we introduce two calculation modes by following Maharana et al. (2023), namely the supervised mode and the unsupervised mode.

*The supervised mode.* We initially train a surrogate model on the dataset. Subsequently, we employ it to calculate sample-wise scores, such as the loss value (Cody Coleman et al., 2019) and gradient norm (Paul et al., 2018). Additionally, we use features before the classification layer as features $\mathbf{F}^{N \times d}$, where $d$ is the feature dimension. We utilize the inner product as the pairwise feature similarity, that is, $\mathbf{K} = \mathbf{F}^{\mathrm{T}} \mathbf{F}$. However, the supervised mode is somewhat unfriendly as training an additional model incurs a non-negligible expense, especially on large-scale datasets.

*The unsupervised mode.* We extract features using existing open-source models such as DINO (Oquab et al., 2023). Additionally, we use the inner product as the pairwise feature similarity. For the intra-sample information, we employ the SSP score (Sorscher et al., 2022). Specifically, it entails first conducting clustering in the feature space. The distance between a sample and the corresponding cluster center constitutes the SSP score.

**Analysis** If we only preserve the first-order term by setting $\alpha$ as a very small value that is near zero, InfoMax will degenerate into a vanilla score-based scheme, that is, those samples with the highest scores will be selected. If we discard the first-order term by setting $\alpha$ as a very huge value, it would degenerate into the problem of finding a coreset $\mathbf{S} \in \mathbf{D}$ to minimize $\sum_{z,s \in \mathbf{S}} \mathbf{K}_{z,s}$. In other words, it is to find a subset with the minimum similarity within the set. This is a variant case of the classic k-Median problem (Lloyd, 1982; Tan et al., 2006; Har-Peled & Mazumdar, 2004), that minimizing the $\mathrm{Cost}(\mathbf{S}, \mathbf{D}) = \sum_{s \in \mathbf{S}} \sum_{z \in \mathbf{D}/\mathbf{S}} d(z, s)$ if we set the distance measurement as $d(z, s) = 1 - \mathbf{K}_{z,s}$.

### 3.2 SALABLE INFOMAX SOLVER

Solving the quadratic problem defined in Eq .2 directly can be extremely computationally burdensome and may even prove intractable since the budget size $p$ is generally on the order of tens of thousands or even millions (Krähenbühl & Koltun, 2011; Larsson et al., 2018). Here, we propose a continuous relaxation of the problem, enabling the use of a gradient-based solver for more efficient optimization. Firstly, we conduct convex relaxation on the feasible domain of the original optimization problem by relaxing the binary constraint $\mathbf{X} \in \{0, 1\}^N$ to a continuous version $\mathbf{X} \in [0, 1]^N$. Then, we derive a solver based on the proximal gradient method (Baqué et al., 2016; Tan et al., 2021). This solver decomposes the original complex and non-convex continuous problem into a series of simple and convex sub-problems, see Appendix D for details. Each sub-problem has an analytic solution that serves as the update rule of our InfoMax solver as follows:

$$\mathbf{X}^{t+1} = \text{Softmax}\left(p \cdot \mathbf{I} - 2p\alpha \cdot \mathbf{K}\mathbf{X}^t\right), \tag{3}$$

where $\mathbf{I} \in \mathbb{R}^N$ is the vectorized $I(z)$ that $\mathbf{I}_z = I(z)$, and $\mathbf{K} \in \mathbb{R}^{N \times N}$ is the similarity matrix. Softmax is the operation to map the real-valued vector $\widetilde{\mathbf{X}}$ to a non-negative vector with the summation of 1, specifically, $\text{Softmax}(\mathbf{X}) = \exp(\mathbf{X})/\sum_i \exp(\mathbf{X})_i$. After $T$ iterations, we get the final solution $\mathbf{X}^T$, of which each item represents the probability that the corresponding sample should be selected. Finally, we take the coreset corresponding to the top-$p$ largest values.

We have summarized InfoMax in Algorithm 1. First, the algorithm requires the input of the intra-sample information vector $\mathbf{I} \in \mathbb{R}^N$ and the pairwise feature similarity $\mathbf{K} \in \mathbb{R}^{N \times N}$. Then, it will iteratively execute the iterative solver defined in Eq. (3) for $T$ iterations. Next, we present additional design enhancements to further improve efficiency.

**Efficiency Enhancement Technique**   Naively applying the InfoMax solver in Eq. (3) on the original dataset leads to a quadratic complexity of $\mathcal{O}(N^2)$, where $N$ is the number of samples to be processed. This complexity is rather high when dealing with large-scale datasets. Here, we introduce two practical techniques to boost the efficiency of InfoMax.

*Dataset partition:* Before executing the InfoMax algorithm, we divide the original dataset into $d$ smaller random subsets and then conduct pruning on each subgroup independently. With this scheme, the complexity of the algorithm on each subset is significantly reduced to $\mathcal{O}(N^2/d^2)$. At the same time, pruning for each subset can be carried out simultaneously on multiple computing devices, further reducing the time consumption.

*Sparsification:* Since the information overlap between distant samples is minimal, we further improve computational efficiency by sparsifying the similarity matrix $\mathbf{K}$, retaining only the top $k$ values as non-zero and setting the rest to zero. Specifically, we only take into account the similarity between each sample and its $k$ nearest neighbors (for instance, $k = 5$). As a result, $\mathbf{K}$ has only $Nk$ non-zero elements. With the sparsification technique, the complexity of the algorithm on each subset is significantly reduced to $\mathcal{O}(Nk/d^2)$.

Note that the partition factor $d$ and the sparsification rate $k$ are two hyperparameters that determine the trade-off between efficiency and performance. In experiments (Sec. 4.4), we also study the effects of these two hyperparameters.

### 3.3 HOW DOES INFOMAX FIND THE MOST INFORMATIVE CORESET?

We explain why InfoMax can find the most informative coreset from the perspective of information theory. First, we restate the information maximization formulation of data pruning defined in Eq. (1), $\mathbf{S}^* = \arg\max_{\mathbf{S} \subset D, |\mathbf{S}|=p} I(\mathbf{S})$, where $p$ is the size of the target coreset. The $I(\mathbf{S})$ measures the set-level information of the candidate subset $\mathbf{S}$, specifically,

$$I(\mathbf{S}) = I(z_1) + I(z_2|z_1) + ... + I(z_p|z_1, ..., z_{p-1}) = I(z_1) + \sum_{2 \leqslant k}^{p} I(z_k|z_1, ..., z_{k-1}). \tag{4}$$

where $\{z_1, ..., z_p\} \in \mathbf{S}$ are all samples from the set. Note that this equation always holds regardless of the order of samples. The intra-sample information $I(z_1)$ of the sample $z_1$ could be instantiated by various sample-wise scores (Sorscher et al., 2022; Cody Coleman et al., 2019; Tan et al., 2023;

Paul et al., 2018). Regarding the conditional gain term $I(z|Z)$, we refer to the recent progress in submodular information measures (SIM) Wei et al. (2015); Kaushal et al. (2021); Kothawade et al. (2021), which present several instantiations for the submodular conditional gain. Here, we select the simplest yet effective instantiation, Graph Cut conditional gain (GCCG), $I(z_k|z_1, ..., z_{k-1}) = f(z_k) - 2\lambda \sum_{i<k} \mathbf{K}_{z_i, z_k}$. It measures the dissimilarity between the sample $z_k$ and all conditional samples. Specifically, $\mathbf{K}_{z_i, z_k}$ measures the similarity between the sample $i$ and the sample $k$, $\lambda$ is an undetermined coefficient and is a hyperparameter in the system. The submodular function $f$ maps from a ground set to a real value, and we can simply set it with $f(z) = I(z)$. Hence, we have the following instantiation for the conditional gain term: $I(z_k|z_1, ..., z_{k-1}) = I(z_k) - 2\lambda \sum_{i<k} \mathbf{K}_{z_i, z_k}$. By substituting it into the Eq. (4), we can instantiate and reformulate the set-level information as: $I(\mathbf{S}) = \sum_{z\in\mathbf{D}} I(z) - 2\lambda \sum_{z\neq s\in\mathbf{D}} \mathbf{K}_{z,s}$. We introduce a set of binary variables $\mathbf{X} \in \{0,1\}^N$ where $N = |\mathbf{D}|$ is the size of the whole training set. In the selection procedure, $\mathbf{X}_z = 1$ indicates the sample $z$ was selected, otherwise it was pruned. By the problem definition in Eq. (1) and the set-level information formulation Eq. (4), we can transform the original information maximization problem in Eq. (1) into the following combinatorial optimization problem,

$$\max_{\mathbf{X}\in\{0,1\}^N, |\mathbf{X}|=p} : \sum_{z\in\mathbf{D}} I(z)\cdot\mathbf{X}_z - \frac{\lambda}{p-1}\sum_{z\neq s\in\mathbf{D}} \mathbf{K}_{z,s}\cdot\mathbf{X}_z\mathbf{X}_s, \tag{5}$$

where $\lambda$ is a flexible hyperparameter to be determined in Eq. (4). Hence, if we set $\alpha = \frac{\lambda}{p-1}$, then we obtain the quadratic programming problem as defined in InfoMax. Therefore, we have proved that under the premise of using the instantiation of Graph-cut conditional gain (GCCG) for the conditional gain term, solving the data pruning problem in Eq. (1) to find the most informative subset is equivalent to solving the quadratic problem defined in Eq. (2). See Appendix D.2 for the proof.

## 4 EXPERIMENTS

We carried out extensive experiments on three tasks, namely image classification, multi-modality pretraining, and instruction tuning for Large Language Models (LLMs), to investigate the performance of our InfoMax. Subsequently, we conducted ablation studies to explore the component design within InfoMax. *Each result of InfoMax is the average of five independent runs. The standard deviation corresponding to each result of InfoMax is less than 0.85.*

### 4.1 IMAGE CLASSIFICATION

The image classification task encompasses experiments on three datasets, namely CIFAR-10, CIFAR-100 (Krizhevsky, 2009), and Imagenet-1K (Russakovsky et al., 2015). Following coreset selection, we will train a model on the chosen subset to examine its performance as the performance of the coreset. The model employed here is ResNet-18 for CIFAR and ResNet-34 for ImageNet. For InfoMax, the dataset partition scheme is not employed herein as the dataset scale is not large. The sparse-rate $k$ is set to 5, the pairwise term weight $\alpha$ is set as 0.3, and the iteration $T$ is set as 20. Regarding the detailed experimental settings, please refer to the Appendix.

For the *supervised* setting, we compare InfoMax with several baselines: 1) **Random** selection of examples. 2) **Entropy** (Cody Coleman et al., 2019) of the model's prediction. 3) **Forgetting** (Toneva et al., 2018) score for each example i.e., the number of times a model predicts the example incorrectly after having predicted correctly in the previous epoch. 4) **EL2N** (Paul et al., 2018), the L2 norm of error vectors. 5) **Moderate** coresets (Xia et al., 2023) that selects samples at about the median distance from the class center, 6) **CCS** (CCS) divides a range of difficulty scores into equal-sized bins and randomly samples from each bin. 7) **$\mathbf{D^2}$-Pruning** (Maharana et al., 2023) selects samples by performing message passing on the data graph. The scoring model for each method is a ResNet model trained on the target dataset. **K-center** (Sener & Savarese, 2017) the standard geometry-based coreset method.

For the *Unsupervised* setting, we select the following baseline: 1) **SSP** (Sorscher et al., 2022) that uses self-supervised embeddings to compute k-means clusters and treats samples at a farther distance from the cluster center as more important, 2) **CCS over SSP scores**, and 3) $D^2$**-Pruning over SSP scores** coreset selection. The unsupervised feature used here is from the officially public DINO-ViT-Base-V2 model (Oquab et al., 2023).

Table 1: A comparative analysis of the performance ((Acc@1)) of InfoMax on image classification datasets with ResNet-18 (on CIFAR) and ResNet-34 (on ImageNet). The best results are bolded.

| Dataset | CIFAR-10 | | | | | | CIFAR-100 | | | | | | ImageNet-1K | | | | | |
|---|---|---|---|---|---|---|---|---|---|---|---|---|---|---|---|---|---|---|
| Selection Rate | 100% | 70% | 50% | 30% | 20% | 10% | 100% | 70% | 50% | 30% | 20% | 10% | 100% | 70% | 50% | 30% | 20% | 10% |
| Random | 95.5 | 94.3 | 93.4 | 90.9 | 88.0 | 79.0 | 78.7 | 74.6 | 71.1 | 65.3 | 57.4 | 44.8 | 73.1 | 72.2 | 70.3 | 66.7 | 62.5 | 52.3 |
| Entropy | - | 94.8 | 92.9 | 90.1 | 84.1 | 72.1 | - | 74.7 | 68.9 | 60.3 | 49.6 | 35.0 | - | 72.3 | 70.8 | 64.0 | 55.8 | 39.0 |
| K-center | - | 94.1 | 92.5 | 91.0 | 82.5 | 68.4 | - | 73.4 | 69.5 | 61.4 | 47.2 | 40.5 | - | 73.0 | 71.4 | 64.8 | 53.1 | 42.0 |
| Forgetting | - | 95.7 | 94.9 | 88.1 | 73.8 | 46.3 | - | 76.0 | 68.1 | 49.3 | 30.3 | 20.6 | - | 72.6 | 70.9 | 66.5 | 62.9 | 52.3 |
| EL2N | - | 95.4 | 94.8 | 89.2 | 78.6 | 30.3 | - | 75.6 | 68.1 | 47.2 | 24.8 | 11.8 | - | 72.2 | 67.2 | 48.8 | 31.2 | 12.9 |
| SSP | - | 93.8 | 93.1 | 81.2 | 64.4 | 42.9 | - | 76.5 | 66.4 | 50.3 | 27.9 | 19.4 | - | 71.1 | 68.5 | 52.7 | 40.3 | 20.5 |
| Moderate | - | 93.9 | 92.6 | 90.6 | 87.3 | 81.0 | - | 74.6 | 71.1 | 65.3 | 58.5 | 45.5 | - | 72.0 | 70.3 | 65.9 | 61.3 | 52.1 |
| CCS (unsupervised) | - | 95.2 | 93.4 | 90.6 | 85.5 | 80.6 | - | 76.4 | 71.8 | 61.2 | 37.5 | 25.4 | - | 71.6 | 69.5 | 62.1 | 47.4 | 30.2 |
| CCS | - | 95.4 | **95.0** | 93.0 | 91.0 | 86.9 | - | 77.1 | 74.4 | 68.9 | 64.0 | 57.3 | - | 72.3 | 70.5 | 67.8 | 64.5 | 57.3 |
| $D^2$-Pruning (unsupervised) | - | 94.4 | 94.2 | 87.6 | 86.1 | 83.9 | - | 77.8 | 70.0 | 66.6 | 62.3 | 51.7 | - | 72.6 | 69.4 | 66.1 | 60.9 | 52.5 |
| $D^2$-Pruning | - | **95.7** | 94.9 | 93.3 | 91.4 | 87.1 | - | 78.2 | 75.9 | 70.5 | 65.2 | 56.9 | - | 72.9 | 71.8 | 68.1 | 65.9 | 55.6 |
| InfoMax (unsupervised) | - | 94.9 | 93.6 | 92.2 | 90.1 | 87.0 | - | 77.9 | 74.6 | 70.1 | 64.7 | 56.2 | - | 72.8 | 70.5 | 68.0 | 64.9 | 56.8 |
| InfoMax | - | 95.5 | 94.7 | **94.1** | **92.7** | **89.1** | - | **79.0** | **76.7** | **71.5** | **67.9** | **58.7** | - | **73.3** | **72.8** | **69.4** | **66.5** | **59.0** |

Table 1 presents the results on three image classification datasets comparing the performance (accuracy) of InfoMax with several baselines. Firstly, let's focus on the performance in the supervised setting. On CIFAR-10, at various pruning rates (30% - 90%), InfoMax outperforms other methods in terms of accuracy in most settings. For instance, at a 90% pruning rate, InfoMax achieves an accuracy of 89.1, outperforming other methods (such as $D^2$-Pruning) by 2.0% in accuracy. On CIFAR-100 and ImageNet, InfoMax has higher accuracy values compared to other methods across different pruning rates. Moreover, this advantage becomes more pronounced under a high pruning ratio. For example, at a 90% pruning rate, InfoMax surpasses the second-ranked CCS by 1.4% and 1.7% on the two datasets respectively, and outperforms the third-ranked $D^2$-Pruning by 1.8% and 3.4% on the two datasets respectively. Next, we must highlight the performance of InfoMax under the unsupervised setting. The performance of InfoMax in the unsupervised setting steadily exceeds that of other schemes in the unsupervised setting. Even under the setting of a high pruning ratio, it can approach and even outperform most supervised schemes. For example, at a 90% pruning rate, the unsupervised InfoMax on CIFAR-10 lags only 0.1% in performance compared with the supervised $D^2$-Pruning, and the unsupervised InfoMax on ImageNet leads the supervised $D^2$-Pruning by a performance advantage of 1.2%. It showcases that InfoMax exhibits superior performance and notable effectiveness across different settings.

## 4.2 MULTI-MODALITY PRE-TRAINING

For multi-modality pretraining tasks, we conducted experiments on the popular vision-language dataset CC12M (Changpinyo et al., 2021), which contains 12 million image-text pairs from the Internet, for CLIP-like vision-language pre-training (Radford et al., 2021). A common practice (Schuhmann et al., 2022; Gadre et al., 2024) for selecting coreset from VL datasets is to use the pre-trained CLIP model (Radford et al., 2021) to score each image-text pair, where a higher CLIP score indicates better image-text alignment. Hence, we set this as the most basic baseline, termed, 1) **CLIP score**. Additionally, we also select the following baseline: 2) **Clustering + CLIP**: Since using only the score for screening is based on the matching degree of images and texts it is difficult to reflect the redundancy degree of samples. Consequently, some schemes (Li et al., 2022; 2023) combine Clustering and CLIP scores, that is, first clustering in the feature space and then selecting a portion of samples with high scores within each cluster. 3) **Moderate** coreset (Xia et al., 2023) over CLIP scores: Selecting those samples with the median score since the highly-scored samples may be the too-easy samples. 4) **CCS** over CLIP scores: By following the implementation in Maharana et al. (2023), it divides a range of CLIP scores into equal-sized bins and randomly samples from each bin. 5) **$D^2$-Pruning** over CLIP scores: It constructs a sample graph by using the CLIP score as the node value and using image features from the CLIP vision encoder to calculate the edge value (similarity).

For our InfoMax approach, we also designate the CLIP score as the intra-sample information measurement. For the pairwise similarity, we employ the inner product between the features of samples from the CLIP vision encoder. The dataset partition factor $d$ is set to 10; that is, we randomly partition the 12M data into 10 subsets each of size 1.2M. The sparse-rate $k$ is set at 5, the pairwise term weight $\alpha$ is set as 0.3, and the iteration $T$ is set to 20. After data selection, we perform CLIP pre-training on the coreset and evaluate the trained model on four downstream tasks: Zero-shot ImageNet-1K Classification, Linear Probing ImageNet-1K Classification, Image-to-Text

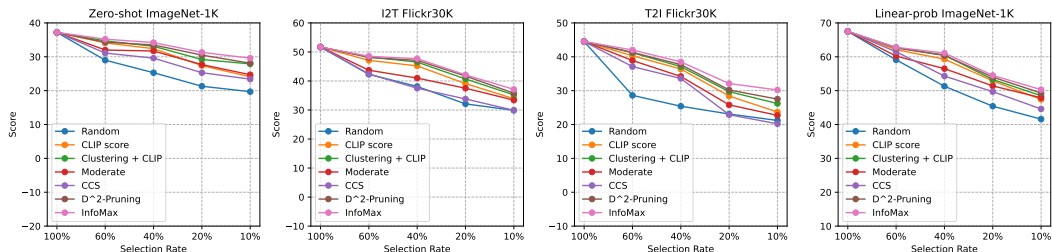

Figure 5: Experimental results of coreset selection on CC12M (Changpinyo et al., 2021) for multi-modality (vision-language) pre-training on CLIP-ViT-B/32 (Radford et al., 2021) model.

(I2T) Flickr30K (Plummer et al., 2015) Retrieval and Text-to-Image (T2I) Flickr30K Retrieval. For the detailed experimental settings and results along with standard errors, please refer to the Appendix.

The experimental results are presented in Table 5. InfoMax consistently outperforms all baselines across all selection ratios. This leading advantage is particularly prominent when the selection ratio is relatively small. For instance, when selection ratios = 10%, InfoMax surpasses $D^2$-Pruning by 1.5%, 1.3%, 2.7%, and 1.0% respectively on these four downstream tasks, and exceeds the widely-used selection method of Clustering + CLIP by 2.1%, 1.9%, 4.0%, and 1.8% respectively on the four tasks. This demonstrates the superiority and effectiveness of InfoMax in different scenarios and under various selection conditions, highlighting its significance in data-centric research areas.

## 4.3 INSTRUCTION TUNING FOR LARGE LANGUAGE MODELS

Recently, some works (Xia et al., 2024; Maharana et al., 2023) have demonstrated significant redundancy in language datasets. For instance, LESS (Xia et al., 2024) reduced the size of instruction tuning datasets to merely 5% of their original amount. The core of LESS is the influence score (Pruthi et al., 2020; Tan et al., 2023), which gauges how a particular data point impacts the performance of the learned model on the validation set.

Following (Xia et al., 2024), we also conduct coreset selection on a mixed dataset containing data from FLAN-V2 (Longpre et al., 2023), COT (Wei et al., 2022), DOLLY (Conover et al., 2023), OPEN-ASSISTANT-1 (Köpf et al., 2023). The mixed training set, consisting of approximately 270K data points, exhibits a wide variation in its format and the underlying reasoning tasks. After the selection process, we conduct LoRA-finetuning (Hu et al., 2021) on the selected coreset for a LLaMA2-7B model (Touvron et al., 2023). For $D^2$-Pruning (Maharana et al., 2023) and our InfoMax, we employ the LESS score as the intra-sample measurement and utilize the gradient of the sample as the sample's feature. Additionally, we select Moderate (Xia et al., 2023) over LESS and CCS (Zheng et al., 2022) over LESS as the baselines.

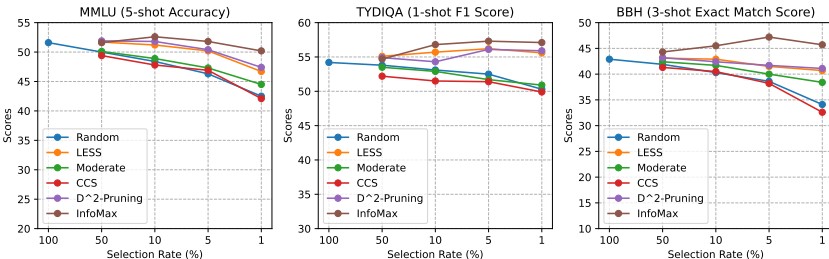

Figure 6: A comparative analysis of the performance of InfoMax on instruction tuning datasets for LLaMA2-7B (Touvron et al., 2023) model.

After the training process, we assess our method using three widely recognized benchmarks: 1) MMLU (Hendrycks et al., 2020) offers a diverse range of knowledge domains for evaluation, covering 57 knowledge areas, like mathematics, computer science, and others. 2) TYDIQA (Clark et al., 2020) is a multilingual question-answer dataset that includes 9 kinds of languages. 3) BBH (Suzgun et al., 2023) encompasses 27 arduous tasks to assess whether the model can handle complex reasoning

situations. The experimental results are presented in Figure 6. InfoMax, our proposed method, demonstrates significant performance advantages over other competing methods, particularly at small selection rates ($\leqslant 5\%$). On MMLU, it shows notable improvements compared to the second-ranked method, $D^2$-Pruning. For example, at the 95% and 99% pruning rates, InfoMax achieves an accuracy of 51.8 and 50.2, compared to $D^2$-Pruning's performance, an improvement of about 2% in performance. Similar trends are observed on TYDIQA, InfoMax outperforms LESS (the second-ranked approach) at 90%, 95%, and 99% pruning rates, with differences ranging from about 1.1 to 1.5 percentage points. For the BBH dataset, the superiority of InfoMax is even more pronounced. At a 95% pruning rate, it has an accuracy of 47.2 compared to $D^2$-Pruning's 41.7, a substantial improvement of about 5.5 percentage points. Overall, InfoMax consistently shows better results, highlighting its effectiveness even when a large portion of the data is pruned.

## 4.4 ANALYSIS & DISCUSSION

Here, we have investigated four hyperparameters in InfoMax: the dataset partition rate $d$, the sparse rate $k$, the weight $\alpha$ of the pairwise term, and the iteration $T$ for InfoMax. In Appendix Sec. B, we will study the time cost and generalization ability of InfoMax. **(a) Partition strategy.** The experiments on CC3M/CC12M (Changpinyo et al., 2021) in Figure 7(a) studies the effect of the dataset partition strategy. It suggests that a decrease in the partitioned subset size inevitably reduces the performance of the coreset. This decline exhibits a significant downward tendency when the size of each subset is less than approximately 1M. Consequently, when dealing with relatively large-scale datasets, we recommend ensuring that the size of each subset is at least 1M samples. **(b) Sparse rate $k$:** The sparse rate $k$ represents the size of the neighborhood of each sample. For details, see Sec. 3.2. A smaller $k$ implies more 0 items in the similarity matrix $K$. In Figure 7(b), we found that the performance of InfoMax is relatively robust to the choice of $k$. Hence, we set $k = 5$ for better efficiency in all experiments. **(c) Pairwise weight $\alpha$:** The pairwise weight $\alpha$ determines the tradeoff between the first-order term and the second-order term in the formulation of InfoMax as in Eq.2. As shown in Figure 7(c), it is observed that the optimal range of $\alpha$ varies for different selection ratios. However, all these ranges yield satisfactory results when $\alpha$ is around 0.3 to 4. Consequently, we recommend setting $\alpha = 0.3$. **(d) Iteration $T$:** Figure 7(d) reveals that the performance of InfoMax rises with the increase in the iteration number $T$. Nevertheless, this gain tends to saturate after $T > 20$. Hence, we suggest setting $T = 20$ to balance good performance and efficiency.

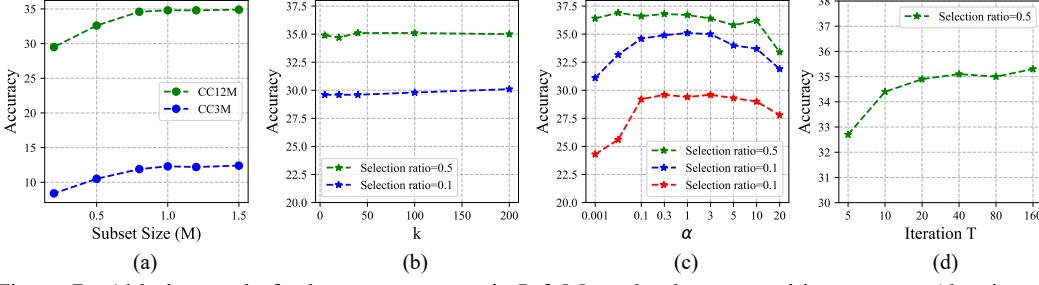

Figure 7: Ablation study for hyperparameters in InfoMax: the dataset partition strategy (the size of each subset), the sparse rate $k$, the weight $\alpha$ of the pairwise term, and the iteration number $T$ of the InfoMax solver. Experiments in (a) are conducted on CC3M & CC12M (Changpinyo et al., 2021). All experiments in (b,c,d) are conducted on CC12M. The selected model is CLIP-ViT-B/32 (Radford et al., 2021). The reported metric focuses on the accuracy of the coreset-trained CLIP-ViT-B/32 on the zero-shot ImageNet-1K classification tasks.

## 5 CONCLUSION

This paper has presented InfoMax, a novel and effective data pruning method. InfoMax is formulated as a quadratic optimization problem to maximize the sample-wise informativeness and minimize the redundancy of selected samples. Furthermore, an efficient gradient-based solver along with sparsification techniques and dataset partitioning strategies is introduced to ensure scalability, enabling it to handle datasets with millions of samples within tens of minutes. Overall, InfoMax shows great potential and effectiveness in the field of data pruning to improve the efficiency and performance of data processing in various applications.

ACKNOWLEDGMENT

This work has been supported in part by the Hong Kong Research Grant Council - Early Career Scheme (Grant No. 27209621), General Research Fund Scheme (Grant No. 17202422, 17212923), Theme-based Research (Grant No. T45-701/22-R), and the Shenzhen Science and Technology Innovation Commission (SGDX20220530111405040). Part of the described research work is conducted in the JC STEM Lab of Robotics for Soft Materials funded by The Hong Kong Jockey Club Charities Trust.

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

BROADER IMPACT

This paper presents a novel and effective data pruning algorithm to advance the deep learning area. There are some potential positive societal effects, such as helping people better understand the role of data to develop more robust deep learning systems and possibly even be used to reduce training and storage costs. Additionally, with the explosive growth of data, similar data-centric deep learning schemes can also be easily applied to inhumane surveillance or scenarios that violate data privacy and data copyright. Therefore, we believe that strict legislation is needed to constrain the occurrence of these.

LIMITATIONS

This paper proposes an efficient and high-performance coreset selection scheme. However, due to the limitations of experimental equipment, the maximum scale of the experiments in this paper only reaches a data scale of 12M. In the future, we will consider investing more in renting or purchasing more computing equipment and exploring the performance boundaries of InfoMax on a larger scale (at the billion level) dataset.

Table 2: Comprehensive overview of the notational convention.

| Notation | Description |
|---|---|
| $\mathbf{D}$ | The training set, and the size of the training set is $|\mathbf{D}| = N$. |
| $\mathbf{S} \subset \mathbf{D}$ | A subset from $\mathbf{D}$. |
| $\mathbf{S}^* \subset \mathbf{D}$ | The optimal coreset. |
| $p$ | The coreset budget and the target coreset size. |
| $z \in \mathbf{D}$ | A training data. |
| $I(z)$ | The information of sample $z$. |
| $I(\mathbf{S})$ | The set-level information of the set $\mathbf{S}$. |
| $I(z|...)$ | The conditional information gain of sample z. |
| $I(z_1, ..., z_n)$ | The set-level information of the set $[z_1, ..., z_n]$. |
| $I(z; s)$ | The mutual information between sample $z$ and $s$. |
| $\mathbf{I} \in \mathbb{R}^N$ | The information vector, where each item measures the informativeness of the corresponding sample, that is, $\mathbf{I}_z = I(z)$. |
| $\mathbf{K} \in \mathbb{R}^{N \times N}$ | The similarity matrix measures the similarity between samples. |
| $\mathbf{K}_{z,s}$ | The similarity between sample $z$ and sample $s$. |
| $k$ | The sparse rate of $\mathbf{K}$. |
| $d$ | The dataset partitioning rate. |
| $\alpha$ | The pairwise weight in the objective of InfoMax, see Eq. (2). |
| $\lambda$ | The hyper-parameter in the Graph-Cut Conditional Gain measurement (Kothawade et al., 2021), see Eq. (15). |
| $\mathbf{X} \in \{0, 1\}^N$ | The binary variable in Eq 2 (before continuous relaxation). |
| $\mathbf{X}_z$ | The variable item corresponding to the sample z. |
| $\mathbf{X}^t \in \mathbb{R}_+^N$ | The iterant variable in the $t$-th iteration (after continuous relaxation). |
| $T$ | The maximum iteration number of the InfoMax solver. |

---

**Algorithm 1:** InfoMax Coreset Selection.

---

1: **Input:** A dataset $\mathbf{D}$ with $N$ samples, the division coefficient $d$, the target coreset size budget $p$.
2: **Initialization:** Set the coreset as a null-set $\mathbf{S}^* = \varnothing$; Divide the dataset $\mathbf{D}$ into $d$ random subsets $[\mathbf{D}_1, ..., \mathbf{D}_d]$, where each subset is size of $N/d$.
3: **Initialization:** Uniformly initialize the initial guess $\mathbf{X}^0 = [\frac{1}{N}]^{N/d}$;
4: **for** $i \in \{1, ..., d\}$ **do**
5:     // Objective construction: supervised or unsupervised mode.
6:     Calculate the intra-information information vector $\mathbf{I}$ and the similarity matrix $\mathbf{K}$;
7:     // Iteratively perform the InfoMax solver.
8:     **for** $t \in \{0, ..., T-1\}$ **do**
9:         $\mathbf{X}^{t+1} = \text{Softmax}\left(p \cdot \mathbf{I} - 2p\alpha \cdot \mathbf{K}\mathbf{X}^t\right)$;
10:     **end for**
11:     Append the samples corresponding to the top-k largest item in $\mathbf{X}^T$ into $\mathbf{S}^*$.
12: **end for**
13: **Output:** The InfoMax coreset $\mathbf{S}^*$.

---

# A    EXPERIMENTAL SETTINGS

## A.1    IMAGE CLASSIICATION

We utilized Pytorch Paszke et al. (2017) to implement our method. Our experiments were conducted on a server equipped with 8 Tesla-V100 GPUs. Additionally, we maintained identical hyper-parameters and experimental settings for training before and after dataset pruning. To ensure fairness, we made sure that the number of iterations on the selected subset and the full set was the same by following Zheng et al. (2022); Maharana et al. (2023). For CIFAR-100, we utilize the SGD optimizer with weight-decay set to 5e-4, a learning rate of 0.1, and a batch size of 128. For TinyImageNet, we use the SGD optimizer with weight-decay set to 5e-4, a learning rate of 0.3, and a batch size of 64. For ImageNet-1K, we use the SGD optimizer with weight-decay set to 1e-4, warmup for 5 epochs, a learning rate of 0.4, and a batch size of 256. Regarding data augmentation, we solely adopt RandomResizedCrop and RandomHorizontalFlip for all experiments.

## A.2    VISION-LANGUAGE PRETRAINING

For coreset selection on the vision-language dataset CC12M (Changpinyo et al., 2021), all experiments are conducted on 2 servers with a totally of 16 NVIDIA V100 GPUs. The selected model is CLIP model (Radford et al., 2021). We follow the settings provided in the original paper. Specifically, the CLIP model Radford et al. (2021) is trained for 32 epochs with AdamW optimizer, weight decay 0.2, and a batch size of 2048. After 1 warmup epoch, the learning rate gradually decreases from 1e-4 following the cosine strategy.

*Zero-shot ImageNet classification.* The CLIP model has two encoders, one for text and one for images. During the zero-shot classification process, text descriptions corresponding to the ImageNet classes are formulated. For example, if a class "dog" exists, a text description like "a picture of a dog" might be created. These text descriptions are encoded by the text encoder of CLIP to obtain text embeddings. At the same time, the images from the ImageNet dataset are encoded by the image encoder of CLIP to get image embeddings. Then, the similarity between each image embedding and all the text embeddings (representing different classes) is calculated. The image is classified into the class whose text embedding has the highest similarity to the image embedding.

*Linear Prob.* This is a technique used to evaluate and analyze the performance of a pre-trained model. For the CLIP model, linear probing involves adding a linear layer on top of the pre-trained CLIP model and then training only this linear layer while keeping the rest of the CLIP model's parameters fixed.

*Image-Text Retrieval.* This is a task where the goal is to find the most relevant text description for a given image or find the most relevant image for a given text description. Let us use the Image-to-Text

Retrieval as an example. The image is encoded using the vision encoder. This results in an image embedding that represents the visual features of the image. Then, text documents are also encoded (using the text encoder) to obtain their respective text embeddings. The similarity between the image embedding and all the text embeddings is computed. The text with the highest similarity score is retrieved as the relevant description for the image.

### A.3 INSTRUCTION TUNING

The specific settings for LoRA fine-tuning are as follows: the Lora-rank is 64, bf-16 precision is used, the number of epochs is 4, the Lora-target-modules include q-proj, k-proj, v-proj, o-proj, the learning rate is $1e^{-05}$, the batch size is 8, the gradient accumulation steps is 16, and the AdamW optimizer is used. This experiments is conducted on a server with 8 A100 GPUs. As for the calculation of the LESS score, please refer to (Xia et al., 2024) for details. Before computing the score, it performs LoRA fine-tuning on the LLM on the training dataset and then retains the randomly-projected LoRA gradient corresponding to each sample. The LESS score reflects how well the gradient of a training sample is consistent with the gradient of the target dataset we are concerned about.

## B FURTHER ANALYSIS

### B.1 GENERALIZATION ABILITY TEST

Here, we demonstrate the generalization ability of InfoMax. In the test for cross-model generalization ability, we observe that InfoMax's coreset exhibits superior generalization ability compared to other coresets. Experimental results are reported in Table 3. InfoMax achieves the best performance in the cross-model generalization ability test. Notably, when using unsupervised scores (SSP (Sorscher et al., 2022)) and unsupervised features from DINO (Oquab et al., 2023), InfoMax has better cross-model generalization ability compared to the supervised version. This also indicates the significance of using unsupervised base models to extract features during data selection.

Table 3: Cross model generalization ability test, including the setting of ResNet to SENet, and the setting of ResNet to EfficientNet-B0. The experiments are conducted on CIFAR-10 (Krizhevsky, 2009). InfoMax (unsupervised) uses the unsupervised scores (SSP (Sorscher et al., 2022)) and unsupervised features from DINO (Oquab et al., 2023), while all remaining methods are based on the EL2N score (Paul et al., 2018) and the feature extractor trained on CIFAR-10.

| Dataset | ResNet | | | ResNet to SENet | | | ResNet to ENet-B0 | | |
|---|---|---|---|---|---|---|---|---|---|
| Selection Rate | 50% | 30% | 20% | 50% | 30% | 20% | 50% | 30% | 20% |
| Random | 93.4 | 90.9 | 88.0 | 93.8 | 92.9 | 88.8 | 90.0 | 87.2 | 84.6 |
| Moderate | 92.6 | 90.6 | 87.3 | 91.4 | 88.3 | 85.7 | 87.1 | 86.9 | 82.2 |
| CCS | 95.0 | 93.0 | 91.0 | 92.7 | 89.4 | 88.1 | 90.5 | 86.7 | 84.1 |
| $D^2$-Pruning | 93.3 | 91.4 | 87.1 | 94.3 | 91.4 | 87.1 | 93.3 | 91.4 | 87.1 |
| InfoMax | 94.1 | 92.7 | 89.1 | 94.4 | 93.3 | 89.9 | 92.5 | 90.6 | 87.8 |
| InfoMax (Unsupervised) | 93.6 | 92.2 | 90.1 | 94.4 | 93.8 | 90.5 | 93.1 | 90.8 | 88.4 |

Furthermore, we assess the cross-setting generalization ability by varying the score (Forgetting (Toneva et al., 2018) and Margin (Har-Peled et al., 2007)) and the type of feature (unsupervised DINO features (Oquab et al., 2023), unsupervised VQGAN features (Esser et al., 2020)). We observe that regardless of the configurations, InfoMax can achieve remarkably superior results compared to score-based approaches that only utilize score or schemes that conduct K-Median Coreset solely with features. This thoroughly demonstrates the cross-setting generalization ability of InfoMax.

### B.2 TIME COST TEST

We study the time cost of InfoMax in Table 5. There are two main stages for InfoMax, the first one is the objective construction stage, including inferencing on all data and calculating the similarity matrix $\mathbf{K}$ and calculating the sample-wise score $\mathbf{I}$, and the second stage is the optimizing stage, which iteratively running the solver defined in Eq. (3). It is easy to find that although the computational efficiency of our method is slower than that of score-based schemes, it is still faster than $D^2$-Pruning

Table 4: Cross setting generalization ability test. The experiments are conducted on CIFAR-10 (Krizhevsky, 2009). The pruning ratio is 10%. VQGAN features are obtained from the VQGAN model (Esser et al., 2020).

| Method | Accuracy |
|---|---|
| Forgetting (Score-based) (Toneva et al., 2018) | 46.3 |
| Margin (Score-based) (Har-Peled et al., 2007) | 34.3 |
| Supervised feature (K-Median diversity-based) | 38.9 |
| DINO feature (K-Median diversity-based) (Oquab et al., 2023) | 31.7 |
| VQGAN feature (K-Median diversity-based) (Esser et al., 2020) | 28.2 |
| InfoMax: forgetting + supervised feature | 89.4 |
| InfoMax: margin + supervised feature | 88.0 |
| InfoMax: margin + DINO feature | 85.4 |
| InfoMax: margin + VQGAN feature | 83.7 |

(Maharana et al., 2023). This is because InfoMax does not have a greedy selection process on a per-sample basis. For CC12M (Changpinyo et al., 2021), we divide the dataset into 10 subsets, each with a size of approximately 1.2 million. We run InfoMax in multiple threads on two server nodes with eight GPUs each to process these subsets. The overall time consumption is approximately 37 minutes. Note that this time is significantly longer than that for processing 1 million ImageNet data because multiple processes occupying the common disk I/O slow down the time efficiency. It is not difficult to observe that the main sources of the aforementioned time consumption are the first stage, including the inference process on all the data and the construction process of the similarity matrix $\mathbf{K}$ with the k-NN algorithm. The block of disk IO by multiple operations will further drag down the efficiency. Therefore, using a more advanced K-NN algorithm and a more advanced disk can significantly improve efficiency.

Table 5: A study on the time cost of InfoMax and two selected competitors, $D^2$-Pruning (Maharana et al., 2023) and score-based method (Entropy (Cody Coleman et al., 2019)), for 1 million image data (the training set of ImageNet-1K (Russakovsky et al., 2015)).

| Name | Time cost (min) |
|---|---|
| Stage-1 of InfoMax | 14.6 |
| Stage-2 of InfoMax | 1.7 |
| Overall of InfoMax | 16.1 |
| $D^2$-Pruning (Maharana et al., 2023) | 23 (Maharana et al., 2023) |
| Score-based method (entropy (Cody Coleman et al., 2019)) | 4.1 |

## C  RELATED WORKS

**Score-based methods.**    The score-based techniques are the most popular data selection approaches. The EL2N score (Paul et al., 2018) measures the data difficulty by computing the average of the $\ell_2$-norm error vector from a set of networks. GraNd (Paul et al., 2018) measures the importance by calculating the expectation of the gradient norm. The Forgetting score (Toneva et al., 2018) counts how many times a model changes its prediction from correct to incorrect for each example during the training process. Memorization (Vitaly Feldman & Chiyuan Zhang, 2020) assigns a score to each example based on how much its presence or absence in the training set affects the model's ability to predict it correctly. Diverse ensembles (Kristof Meding et al., 2022) gave a score to each sample based on how many models in a group misclassified it. (Sorscher et al., 2022) proposed to use the distance between the sample and its corresponding cluster center as the importance score. Influence score (Tan et al., 2023; Xia et al., 2024) measures the sample-wise leave-one-out retraining influence on the model's performance. The score-based approach often suffers from performance problems in application, especially when the pruning ratio is large. Some recent works have tried to solve this problem through various methods, for example, Moderate (Xia et al., 2023) suggested selecting data points with scores close to the score median. Note that Moderate (Xia et al., 2023) can use any selection criterion, such as EL2N score (Paul et al., 2018), as a basis. Dyn-Unc He et al. (2024) proposed an efficient uncertainty-based score with awareness of training dynamics. Some related works also use sample-wise scores (Radford et al., 2021; Mahmoud et al., 2024) to reflect the quality of multi-modality data.

**Diversity-based (Geometry-based) methods.** Traditionally, diversity-based coreset schemes are a very classic computer science problem (Lloyd, 1982; Tan et al., 2006; Coates & Ng, 2012; Har-Peled & Mazumdar, 2004; Feldman & Langberg, 2011; Feldman et al., 2013; Jiang et al., 2024). These schemes aim to find a subset of data points that maximizes the diversity among the selected elements. Sener & Savarese (2017) applied greedy k-center to choose the coreset with good data coverage. Yu et al. (2020); Chan et al. (2022) proposed to use the coding rate to model measure the diversity. Yu et al. (2022) formulates the problem of finding the most diverse subset into the problem of maximizing the dispersion or convex hull volume. In addition, some works proposed to prune data from the perspective of submodule theory (Wei et al., 2015; Kaushal et al., 2021; Kothawade et al., 2021) and linear programming (Yang et al., 2023) to ensure diversity.

**Hybrid methods.** Solely using the diversity-based method alone can hardly perform satisfactorily because they do not consider the intra-sample information. Recently, some hybrid works have also attempted to introduce that score-based scheme into the diversity-driven pipelines to achieve better performance. CCS (Zheng et al., 2022) sets different sampling ratios for samples with different scores to enhance data coverage and balance both easy and hard samples. Yang et al. (2024) introduces reconstructing the classification boundary on the original dataset as a goal and brings it into the framework of CCS. BADGE (Jordan T Ash et al., 2019) is a diversity-based selection method in active learning that clusters the gradient embeddings of the current model using k-means and selects a subset from each cluster. $D^2$-Pruning (Maharana et al., 2023) views data pruning as a node selection problem based on Message-Passing on a graph, where the intra-sample information is utilized as the node values on the sample graph. One of the core steps of $D^2$-Pruning is the Inverse Message Passing operation, which iteratively performs a greedy sample selection step. In each iteration, it will select the sample with the highest score from all unselected candidates and then reduce the score of samples in the neighborhood to guarantee that highly redundant parts are not selected in the subsequent iterations. Among the above methods, $D^2$-Pruning is currently the most advanced solution in terms of performance. However, due to the heuristic or greedy nature of the algorithm, the result obtained by using is often suboptimal, check Figure 1 for details.

# D PROOF

## D.1 DERIVATION OF THE INFOMAX SOLVER

Firstly, we restate the quadratic optimization problem of InfoMax:

$$\max_{\mathbf{X} \in \{0,1\}^N} \sum_{z \in \mathbf{D}} \mathbf{X}_z I(z) - \alpha \cdot \sum_{z,s \in \mathbf{D}} \mathbf{K}_{z,s} \mathbf{X}_z \mathbf{X}_s, \quad \text{s.t.} \quad \sum_{z \in \mathbf{D}} \mathbf{X}_z = p, \tag{6}$$

Solving the quadratic problem defined in Eq .2 directly can be extremely computationally burdensome and may even prove intractable since the budget size $p$ is generally on the order of tens of thousands or even millions (Krähenbühl & Koltun, 2011; Larsson et al., 2018). Continuous relaxation simplifies the problem by relaxing some of the discrete constraints to continuous ones, reducing the complexity of the search space and making the problem more amenable to efficient optimization algorithms, such as the gradient-based methods (Krähenbühl & Koltun, 2011; Larsson et al., 2018). Here, we also introduce a continuous relaxation of the problem, enabling the use of a gradient-based solver for more efficient optimization:

$$\max_{\mathbf{X} \in \mathbb{R}_+^N} \sum_{z \in \mathbf{D}} \mathbf{X}_z I(z) - \alpha \cdot \sum_{z,s \in \mathbf{D}} \mathbf{K}_{z,s} \mathbf{X}_z \mathbf{X}_s, \quad \text{s.t.} \quad \sum_{z \in \mathbf{D}} \mathbf{X}_z = p, \tag{7}$$

According to (Krähenbühl & Koltun, 2011; Larsson et al., 2018; Baqué et al., 2016; Tan et al., 2021), the continuous (complex and non-convex) problem in Eq. (7) could be optimized by solving a series of the following (convex) sub-problems:

$$\mathbf{X}^{t+1} = \arg \min_{\mathbf{X} \in \mathbb{R}_+^N, \sum_{z \in \mathbf{D}} \mathbf{X}_z = p} -\mathbf{X}^{\mathrm{T}} \left( \mathbf{I} - 2\alpha \mathbf{K} \mathbf{X}^t \right) + \lambda h(\frac{\mathbf{X}}{p}) + \frac{1}{\beta} D(\mathbf{X}, \mathbf{X}^t), \tag{8}$$

where $X^t$ is the solution of the $t$-th sub-problem, $\left( \mathbf{I} - 2\alpha \mathbf{K} \mathbf{X}^t \right)$ is the gradient of the objective in Eq. (7) at $\mathbf{X}^t$. We introduce the convex entropy function $h(\cdot)$ controlled by a factor $\lambda$ the

regularization term. A large $\lambda$ value makes the problem easier to solve (Krähenbühl & Koltun, 2011; Larsson et al., 2018; Baqué et al., 2016; Tan et al., 2021), but it may deviate from the original problem. The proximal operator $D(\mathbf{X}, \mathbf{X}^t)$ is an optional regularization term to prevent the solution difference between the two iterations is too large. Following tradition (Krähenbühl & Koltun, 2011; Larsson et al., 2018; Baqué et al., 2016; Tan et al., 2021), we use $D(\mathbf{X}, \mathbf{X}^t) = \frac{\mathbf{X}}{p} \log \frac{\mathbf{X}}{p} - \frac{\mathbf{X}}{p} \log \frac{\mathbf{X}^t}{p}$, which is the Kullback-Leibler divergence measure the discrepancy between $\frac{\mathbf{X}}{p}$ and $\frac{\mathbf{X}^t}{p}$. Note that due to the non-negativity of $\mathbf{X}$ and the property that the sum is a fixed value $p$, $\frac{\mathbf{X}}{p}$ has a probabilistic meaning. Each element of it represents the probability that each sample is selected. If we differentiate this convex sub-problem, the optimal solution is identified by solving the following equation:

$$0 = -\left(\mathbf{I} - 2\alpha \mathbf{K}\mathbf{X}^t\right) + \lambda\left(-\frac{1}{p}\log\frac{\mathbf{X}^*}{p} - 1\right) + \frac{1}{\beta}\left(\frac{1}{p}\log(\frac{\mathbf{X}^*}{\mathbf{X}^t}) + \frac{1}{p}\mathbf{X}^t\right), \quad \text{s.t.} \quad \mathbf{X}^* \in \mathbb{R}_+^N, \sum_{z \in \mathbf{D}} \mathbf{X}_z^* = p,$$
(9)

This has an analytic solution to (Krähenbühl & Koltun, 2011; Larsson et al., 2018; Baqué et al., 2016; Tan et al., 2021),

$$\hat{\mathbf{X}}^* = \exp\left(\frac{\beta p}{\lambda \beta - 1}(\mathbf{I} - 2\alpha \mathbf{K}\mathbf{X}^t) + \frac{1}{1 - \lambda\beta}(\log\frac{\mathbf{X}^t}{p} - \mathbf{X}^t) + \frac{\lambda\beta p}{1 - \lambda\beta}\right),$$
(10)

$$\mathbf{X}^{t+1} = \mathbf{X}^* = \hat{\mathbf{X}}^* / (\sum_z \hat{\mathbf{X}}_z^*).$$
(11)

Since the third term in Eq.10 is a constant, the solution could be formulated as the following form:

$$\mathbf{X}^{t+1} = \text{Softmax}\left(\frac{\beta p}{\lambda\beta - 1}(\mathbf{I} - 2\alpha \mathbf{K}\mathbf{X}^t) + \frac{1}{1 - \lambda\beta}(\log\frac{\mathbf{X}^t}{p} - \mathbf{X}^t)\right).$$
(12)

The mapping by the exponential function $\exp$ followed by the summation normalization is just equivalent to the Softmax operator which is widely used in deep learning. By just setting $\lambda = 1$ and setting $\beta \to \infty$, we have the most simplified iterative solution:

$$\mathbf{X}^{t+1} = \text{Softmax}\left(p \cdot \mathbf{I} - 2p \cdot \alpha \mathbf{K}\mathbf{X}^t\right).$$
(13)

Note that the convergence rate of this solver is quite fast. In particular, the norm of the iterant difference $|\mathbf{X}^{t+1} - \mathbf{X}^t|$ converges at a rate of $\mathcal{O}(\frac{1}{T})$ according to (Baqué et al., 2016; Tan et al., 2021).

### D.2   PROD: HOW DOES INFOMAX FIND THE MOST INFORMATIVE CORESET?

We explain why InfoMax can find the most informative coreset from the perspective of information theory. First, we restate the information maximization formulation of data pruning defined in Eq. (1), $\mathbf{S}^* = \arg\max_{\mathbf{S} \subset D, |\mathbf{S}|=p} I(\mathbf{S})$, where $p$ is the size of the target coreset. The $I(\mathbf{S})$ measures the set-level information of the candidate subset $\mathbf{S}$, specifically,

$$I(\mathbf{S}) = I(z_1) + I(z_2|z_1) + ... + I(z_p|z_1, ..., z_{p-1})$$
$$= I(z_1) + \sum_{2 \leqslant k}^p I(z_k|z_1, ..., z_{k-1}).$$
(14)

where $\{z_1, ..., z_p\} \in \mathbf{S}$ are all samples from the set. Note that this equation always holds regardless of the order of samples. The intra-sample information $I(z_1)$ of the sample $z_1$ could be instantiated by various sample-wise scores (Sorscher et al., 2022; Cody Coleman et al., 2019; Tan et al., 2023; Paul et al., 2018). Regarding the conditional gain term $I(z|Z)$, we refer to the recent progress in submodular information measures (SIM) Wei et al. (2015); Kaushal et al. (2021); Kothawade et al. (2021), which present several instantiations for the submodular conditional gain. Here, we select the simplest yet effective instantiation, Graph Cut conditional gain (GCCG),

$$I(z_k|z_1, ..., z_{k-1}) = f(z_k) - 2\lambda \sum_{i<k} \mathbf{K}_{z_i, z_k}.$$
(15)

It measures the dissimilarity between the sample $z_k$ and all conditional samples. Specifically, $\mathbf{K}_{z_i, z_k}$ measures the similarity between the sample $i$ and the sample $k$, $\lambda$ is an undetermined coefficient

and is a hyperparameter in the system. The submodular function $f$ maps from a ground set to a real value, and we can simply set it with $f(z) = I(z)$. Hence, we have the following instantiation for the conditional gain term: $I(z_k|z_1, ..., z_{k-1}) = I(z_k) - 2\lambda \sum_{i<k} \mathbf{K}_{z_i, z_k}$. By substituting it into the Eq. (14), we can instantiate and reformulate the set-level information as:

$$I(\mathbf{S}) = \sum_{z \in \mathbf{D}} I(z) - 2\lambda \sum_{z \neq s \in \mathbf{D}} \mathbf{K}_{z,s}. \tag{16}$$

We introduce a set of binary variables $\mathbf{X} \in \{0,1\}^N$ where $N = |\mathbf{D}|$ is the size of the whole training set. In the selection procedure, $\mathbf{X}_z = 1$ indicates the sample $z$ was selected, otherwise it was pruned. By the problem definition in Eq. (1) and the set-level information formulation Eq. (14), we can transform the original information maximization problem in Eq. (1) into the following combinatorial optimization problem,

$$
\begin{aligned}
\max_{\mathbf{X} \in \{0,1\}^N, |\mathbf{X}|=p} : \quad & \sum_{\mathbf{S} \subset \mathbf{D}} \prod_{z \in \mathbf{S}} \mathbf{X}_z I(\mathbf{S}) \\
= & \frac{1}{p!} \sum_{z_1 \neq ... \neq z_p \in \mathbf{D}} \prod_{i \leqslant p} \mathbf{X}_{z_i} I([z_1, ..., z_n]) \\
= & \frac{1}{p!} \sum_{\mathbf{S} \subset \mathbf{D}} \prod_{z \in \mathbf{S}} \mathbf{X}_z \Big( \sum_{z \in \mathbf{S}} I(z) - 2\lambda \sum_{z \neq s \in \mathbf{S}} \mathbf{K}_{z,s} \Big), \\
= & \frac{1}{p!} \sum_{z \neq z_1 ... \neq z_{p-1} \in \mathbf{D}} \mathbf{X}_z \mathbf{I}_z \prod_{1 \leqslant i}^{p-1} \mathbf{X}_{z_i} - 2\lambda \frac{1}{p!} \sum_{z \neq s \neq z_1 ... \neq z_{p-2} \in \mathbf{D}} \mathbf{X}_z \mathbf{X}_s \mathbf{K}_{z,s} \prod_{1 \leqslant i}^{p-2} \mathbf{X}_{z_i} \\
= & \frac{1}{p!} \sum_{z \neq z_1 ... \neq z_{p-2} \in \mathbf{D}} \mathbf{X}_z \mathbf{I}_z \prod_{1 \leqslant i}^{p-2} \mathbf{X}_{z_i} \Big( \sum_{z_{p-1}} \mathbf{X}_{z_{p-1}} - \sum_{j=1}^{p-2} \mathbf{X}_{z_j} \Big) \\
& - 2\lambda \frac{1}{p!} \sum_{z \neq s \neq z_1 ... \neq z_{p-3} \in \mathbf{D}} \mathbf{X}_z \mathbf{X}_s \mathbf{K}_{z,s} \prod_{1 \leqslant i}^{p-3} \mathbf{X}_{z_i} \Big( \sum_{z_{p-2}} \mathbf{X}_{z_{p-2}} - \sum_{j=1}^{p-3} \mathbf{X}_{z_j} \Big) \\
= & \frac{1}{p!} \sum_{z \neq z_1 ... \neq z_{p-2} \in \mathbf{D}} \mathbf{X}_z \mathbf{I}_z \prod_{1 \leqslant i}^{p-2} \mathbf{X}_{z_i} \Big( p - \sum_{j=1}^{p-2} \mathbf{X}_{z_j} \Big) \\
& - 2\lambda \frac{1}{p!} \sum_{z \neq s \neq z_1 ... \neq z_{p-2} \in \mathbf{D}} \mathbf{X}_z \mathbf{X}_s \mathbf{K}_{z,s} \prod_{1 \leqslant i}^{p-3} \Big( p - \sum_{j=1}^{p-3} \mathbf{X}_{z_j} \Big)
\end{aligned}
\tag{17}
$$

Since $\mathbf{X}$ is binary, hence,

$$
\begin{aligned}
\max_{\mathbf{X} \in \{0,1\}^N, |\mathbf{X}|=p} : \quad & \sum_{\mathbf{S} \subset \mathbf{D}} \prod_{z \in \mathbf{S}} \mathbf{X}_z I(\mathbf{S}) \\
= & \frac{1}{p!} \sum_{z \neq z_1 ... \neq z_{p-2} \in \mathbf{D}} \mathbf{X}_z \mathbf{I}_z \prod_{1 \leqslant i}^{p-2} \mathbf{X}_{z_i} \Big( p - \sum_{j=1}^{p-2} \mathbf{X}_{z_j} \Big) \\
& - 2\lambda \frac{1}{p!} \sum_{z \neq s \neq z_1 ... \neq z_{p-2} \in \mathbf{D}} \mathbf{X}_z \mathbf{X}_s \mathbf{K}_{z,s} \prod_{1 \leqslant i}^{p-3} \mathbf{X}_{z_i} \Big( p - \sum_{j=1}^{p-3} \mathbf{X}_{z_j} \Big) \\
= & \frac{1}{p!} p \cdot \sum_{z \neq z_1 ... \neq z_{p-2} \in \mathbf{D}} \mathbf{X}_z \mathbf{I}_z \prod_{1 \leqslant i}^{p-2} \mathbf{X}_{z_i} - \frac{1}{p!}(p-2) \cdot \sum_{z \neq z_1 ... \neq z_{p-2} \in \mathbf{D}} \mathbf{X}_z \mathbf{I}_z \prod_{1 \leqslant i}^{p-2} \mathbf{X}_{z_i} \\
& - 2p \cdot \lambda \frac{1}{p!} \sum_{z \neq s \neq z_1 ... \neq z_{p-2} \in \mathbf{D}} \mathbf{X}_z \mathbf{X}_s \mathbf{K}_{z,s} \prod_{1 \leqslant i}^{p-3} \mathbf{X}_{z_i} + 2(p-3) \cdot \lambda \frac{1}{p!} \sum_{z \neq s \neq z_1 ... \neq z_{p-2} \in \mathbf{D}} \mathbf{X}_z \mathbf{X}_s \mathbf{K}_{z,s} \prod_{1 \leqslant i}^{p-3} \mathbf{X}_{z_i} \\
= & 2 \frac{1}{p!} \cdot \sum_{z \neq z_1 ... \neq z_{p-2} \in \mathbf{D}} \mathbf{X}_z \mathbf{I}_z \prod_{1 \leqslant i}^{p-2} \mathbf{X}_{z_i} - 6 \cdot \lambda \frac{1}{p!} \sum_{z \neq s \neq z_1 ... \neq z_{p-2} \in \mathbf{D}} \mathbf{X}_z \mathbf{X}_s \mathbf{K}_{z,s} \prod_{1 \leqslant i}^{p-3} \mathbf{X}_{z_i}
\end{aligned}
\tag{18}
$$

By applying a similar reduction process to the rest variables, we have:

$$
\begin{aligned}
\max_{\mathbf{X} \in \{0,1\}^N, |\mathbf{X}|=p} : \quad & \sum_{\mathbf{S} \subset \mathbf{D}} \prod_{z \in \mathbf{S}} \mathbf{X}_z I(\mathbf{S}) \\
= & \frac{1}{p!}(p-1)! \cdot \sum_{z \in \mathbf{D}} \mathbf{X}_z \mathbf{I}_z - (p-2)! \cdot \lambda \sum_{z \neq s \in \mathbf{D}} \mathbf{X}_z \mathbf{X}_s \mathbf{K}_{z,s} \\
= & \frac{(p-1)!}{p!} \Big( \sum_{z \in \mathbf{D}} I(z) \cdot \mathbf{X}_z - \frac{\lambda(p-2)!}{(p-1)!} \sum_{z \neq s \in \mathbf{D}} \mathbf{K}_{z,s} \cdot \mathbf{X}_z \mathbf{X}_s \Big),
\end{aligned}
\tag{19}
$$

where $(p)!$ is a factorial function of $p$, and $\lambda$ is a hyperparameter to be determined in Eq. (4). And we have $\frac{(p-2)!}{(p-1)!} = \frac{1}{p-1}$. Hence, if we use $\alpha$ to indicate $\frac{\lambda(p-2)!}{(p-1)!}$, we obtain the quadratic programming

problem as defined in InfoMax. Therefore, we have proved that under the premise of using the instantiation of Graph-cut conditional gain (GCCG) for the conditional gain term, solving the data pruning problem in Eq. (1) to find the most informative subset is equivalent to solving the quadratic problem defined in Eq. (2).

