# OpenReview forum: "Data Pruning by Information Maximization"
_ICLR.cc/2025/Conference — ICLR 2025 Poster_

### Official Review · Reviewer_2PGZ · 2024-11-02

**Soundness:** 2
**Presentation:** 3
**Contribution:** 2
**Rating:** 5
**Confidence:** 4

**Summary:**

This paper targets at designing a new coreset selection method to enhance the informativeness of the coreset. Specifically, the authors formalize the coreset selection problem as a discrete quadratic programming task. They adopt an objective of indivisual sample contribution minus the redundancies introduced by similar samples within the coreset. An efficient gradient-based solver is introduced to enhance the efficiency for the implementation. The method is proven to be effective on multiple coreset selection benchmarks including both vision and language tasks.

**Strengths:**

1. The proposed informax solver is interesting, and can be a solution for coreset selection optimization.
2. The proposed method achieves state-of-the-art performance on multiple benchmarks including both vision and language data.
3. The presentation of the paper is professional.

**Weaknesses:**

1. The framework of the proposed Infomax method is reasonable. However, the actual implementation simply adopts the existing submodular and graph cut methods. And there is no comparison with graph cut in the experiment section. Can the authors give more detailed explanation what is the difference of the proposed method from graph cut? Will the proposed gradient-based solver improve the performance over the original graph cut?
2. The authors propose an efficiency enhancement technique, where the original dataset is divided into several subsets, and only the similarity between neighbors is calculated.
    - Have the authors compare the efficiency enhancement technique with some existing efficient similarity calculation techniques, e.g., faiss?
    - The technique is not only applicable to the proposed Infomax method, but also to previous coreset selection methods. The authors claim that the proposed method achieves faster selection compared with D$^2$-Pruning. The comparison seems a little bit unfair.
3. While efficiency is a major advantage of the proposed method, the authors only provide one group of time cost comparison. As the computation time will be affected by the coreset size, more comparison is expected under different coreset sizes. Futhermore, there is also expected to be the time comparison w/ or w/o each proposed efficient module.
4. In section 4.4(a), the text part says partition rate d, but in figure 5(a) it says subset size. Although they refer to the same thing, it would still be better to keep these two terms consistent.
5. The original paper surpasses the limit of 10 pages.

**Questions:**

1. The authors adopt Dinov2 to extract embeddings for the unsupervised scenario. Will the model choice matter?
2. The authors apply softmax on $\mathbf{X}$. Will this operation have large influence on the results? How about simple normalize? I cannot see the exact motivation of adopting a softmax here.

---

> ### Author Response · Authors · 2024-11-23
> **Response to Reviewer 2PGZ**
>
> We sincerely appreciate your constructive comments on our work. If you have any additional concerns, please do not hesitate to reach out. We are committed to addressing your feedback. Thank you!
>
> ---
> **Weakness-1: Questions about the relation between InfoMax and submodular/graph-cut methods**
>
> Thanks!
>
> We want to clarify that InfoMax is based on an information perspective. It turns the data pruning problem into a combinatorial optimization problem by looking at both the importance of individual samples and the redundancy between them. We also created an efficient solver for this problem, allowing InfoMax to perform well in various situations.
>
> Section 3.3 explains why InfoMax is focused on finding the most informative subset. This connection is made by using the graph-cut conditional gain from submodular information theory as the basis for each conditional information gain in Eq.(4). Therefore, InfoMax is not just an incremental improvement on graph-cut or submodular theory.
>
>
>
> **Weakness-2: Why not use the FAISS toolkit but just divide the original large set? And is the speed comparison fair?**
>
> Thanks!
>
> First, FAISS can speed up the construction of the K-NN graph. For example, it can create a K-NN graph for a 12 million vision-language dataset within an hour. However, with datasets in the billions, using only FAISS can still take a long time. To tackle this, we introduce a technique called dataset partitioning, which breaks the large dataset into smaller subsets.
>
> Our ablation experiments for the vision-language pretraining task, shown in Figure 5(a), indicate that keeping the subset size above 1 million is enough to maintain performance.
>
> We ensure a fair comparison by using the same partitioning method for other baselines, like D2-Pruning, in both speed and performance tests.
>
> **Weakness-3: Moreover speed comparison on a larger dataset, and different pruning ratio settings.**
>
>
> Thanks!
>
> (a). Pruning ratio and speed. The pruning ratio doesn’t impact the time cost of InfoMax. This is because the iteration variable ${X}$ in the InfoMax solver is a binary vector that matches the size of the data. If $X_i=1$, the sample is included in the coreset; if not, it’s discarded. So, the complexity of the iteration depends only on the amount of data, not the pruning ratio.
>
> (b). Further speed test. Here, we present a speed comparison between InfoMax and D2-Pruning across larger-scale scenarios, including 100 million, and 1 billion samples. The reported metric is GPU hours as shown below..
> For both InfoMax and D2-Pruning, we utilize the same acceleration strategy, that is, randomly partitioning the dataset into subsets with the size of 1M. The K-NN graph is constructed with FAiSS, where k=5.
>
> Method|100M|1000M
> ---|---|---
> D2-Pruning|8.7|78.3
> InfoMax|7.4|75.2
>
> For both InfoMax and D2-Pruning, which require graph construction, we applied the Efficiency Enhancement Techniques described in Section 3.2. Overall, InfoMax outperforms D2-Pruning in speed, as its solving process can be executed rapidly in parallel on a GPU, whereas the greedy selection process of D2-Pruning must be carried out serially.
>
>
>
>
>
> **Weakness-4: In section 4.4(a), the text part says partition rate d, but in figure 5(a) it says subset size. Although they refer to the same thing, it would still be better to keep these two terms consistent.**
>
> Thanks! We have revised section 4.4 according to your suggestion.
>
> **Weakness-5: About the paper length.**
>
> Thanks! We have fixed this mistake.
>
> **Question-1: More feature choices rather than DINO-v2.**
>
> Thanks! In Table 6, we evaluate InfoMax using a different unsupervised feature extractor called VQGAN. We also test InfoMax with BYOL and MSN features on ImageNet-1K, keeping a selection rate of 10\%. We measure sample informativeness using the SSP score, which looks at the distance between each sample and its cluster center in the feature space.
>
> Feature extractor|Top-1 Acc on ImageNet-1K
> ---|---
> EL2N|12.9
> D2-Pruning|55.6
> InfoMax|59.0
> InfoMax with BYOL|50.9
> InfoMax with MSN|54.2
>
>
>
> **Question-2: Why use Softmax in the InfoMax solver?**
>
> Thanks!
>
> We can’t change the softmax normalization in Eq.(9) of the main paper because it comes from a mathematical derivation, and Eq.(9) is the optimal solution for the sub-problem in Eq.(8). We provide the detailed proof in the revised Appendix D.1. In our work, we transform the original non-convex problem into several convex sub-problems, each with a clear solution given by Eq.(9) using the softmax operation. If we used different operations, the update wouldn’t necessarily be the optimal solution for the sub-problem, which could affect the overall convergence of the InfoMax algorithm.

---

> > ### Comment · Reviewer_2PGZ · 2024-11-25
> >
> > Thanks for the reply.
> >
> > From the reply to W1, the main contribution is the efficient solver. But according to the reply to W3, under a fair comparison, Infomax doesn't illustrate a significant reduction in terms of the running time, especially for the 1B case.
> >
> > I think my concern is not fully addressed.

---

> ### Author Response · Authors · 2024-11-23
> **Reference**
>
> [1] BYOL: Bootstrap your own latent: A new approach to self-supervised Learning. NeurIPS 2020.
>
> [2] MSN: Masked Siamese Networks for Label-Efficient Learning. ECCV 2022.
>
> [3] Beyond neural scaling laws: beating power law scaling via data pruning. NeurIPS 2022.
>
> [4]. Submodularity in data subset selection and active learning. ICML 2015.

---

> ### Author Response · Authors · 2024-11-25
> **Looking forward to your further reply**
>
> Dear Reviewer 2PGZ:
>
> We genuinely appreciate your efforts in reviewing our paper and your valuable suggestions for our work. As we near the conclusion of the discussion period, we would like to inquire if there are any further concerns regarding our paper or our response. We are more than willing to address any additional questions you may have.
>
> Best regards,
>
> Authors of InfoMax

---

> ### Author Response · Authors · 2024-11-25
> **Sincerely response to Reviewer 2PGZ**
>
> We truly appreciate the opportunity for further discussion with you.
>
> ---
>
> 1. **Regarding the novelty**, we would like to highlight that our InfoMax approach is fundamentally different from other data pruning methods in both motivation and formulation. We have also provided an information-theoretic perspective on the weaknesses of existing approaches. We believe that in terms of performance, InfoMax consistently achieves the best results.
>
> 2. **We are sure about the fairness:** We would be grateful for your clarification on why you feel the comparison is unfair. We only apply the dataset partitioning technique in experiments involving datasets larger than 12M, and this same technique is also utilized in other methods, such as D2-Pruning. We believe this makes the comparison valid.
>
> 3. **InfoMax can be efficient enough:** Lastly, we would like to point out that InfoMax is indeed faster than D2-Pruning. While other methods, such as score-based approaches, may have quick execution times, their performance can often be lacking. Both InfoMax and D2-Pruning encounter speed limitations during the initial phase of constructing the sample graph, which is a requirement for all graph-based methods. As shown in the figure below, the time required for InfoMax alone can be quite low—just 4.8 hours on 16 GPUs (3090) for 1000M samples.
>
> Method on 1000M Samples|Graph Construction|cost in total | overall-time on 2*8 GPUs
> ---|---|---|---
> D2-Pruning|59 GPU-Hours |78.3 GPU-Hours | 5.7 Hours
> InfoMax|59 GPU-Hours|75.2 GPU-Hours|4.8 Hours
>
> ---
>
> Thank you for considering our points, and we look forward to your insights!
>
>
> Wish you all the best,
>
> Submission1258 Authors

---

> > ### Comment · Reviewer_2PGZ · 2024-11-25
> >
> > Thanks for the prompt reply.
> >
> > 1. I still cannot see the major difference in the method design from submodular and graphcut.
> > 2. The running time in total only reduces 3.1 GPU hours from D2-Pruning (4%). Why on 2*8 GPUs the time difference ratio (15.8%) is larger?

---

> ### Author Response · Authors · 2024-11-25
> **Response to Reviewer 2PGZ**
>
> Thanks for your reply! We are happy to address your concerns!
>
> ---
>
> 1. The only connection between InfoMax and submodular/graphcut is that in Sec.3.3 when we explain why optimizing the quadratic optimization target defined in (2) is equivalent to finding the most informative subset, we use the Graph-cut-conditional gain (GCCG) instantiations in the sub-modular theory [1,2] to proof this equivalence. This has been discussed in lines 1264-1266 in the revision.
>
>    [1] Submodularity in data subset selection and active learning. ICML-2015.
>
>    [2] Similar: Submodular information measures based active learning in realistic scenarios. NeurIPS 2021.
>
> 2. Consider the time cost for each sub-task on each GPU is $t_i, i \in [0,1,...,15]$. Then,  cost-in-total $= \sum_i t_i$, the overall-time-cost$=\max (t_i) $. This can answer your question!
>
> We are more than willing to address any additional questions you may have.

---

> > ### Comment · Reviewer_2PGZ · 2024-11-25
> >
> > 1. Yes, it exactly means that you adopt the implementation of graphcut and submodular.
> > 2. Does the result mean Infomax can lead to more even time distribution across GPUs? But it doesn't reduce the overall GPU hours or the required calculation.

---

> ### Author Response · Authors · 2024-11-25
> **Response to Reviewer 2PGZ**
>
> We sincerely thank you for your reply! Thank you for your time and contribution to the academic community.
>
> ---
>
> Q.1 To address your concerns, we would like to clarify the difference and connection between InfoMax and Graph-cut (specifically, the Graph-cut-conditional gain instantiation in submodular measurement theory):
>
> **a. Difference:** The distinction between InfoMax and Graph-cut/submodular theory is significant.
>
> **InfoMax** is a data pruning method that is based on an information-theoretic perspective for dataset pruning and coreset selection tasks. It provides a unified view by solving a quadratic discrete optimization problem to find the optimal coreset with maximum importance and minimal redundancy. *
>
> **Submodular measurement theory** serves as a theoretical framework that extends information theory, allowing for the measurement of nodes or individuals rather than just variables or distributions. The Graph-cut you mentioned refers to the Graph-cut-conditional gain instantiation, which is a specific measurement instantiation under this theoretical framework.
>
> **b. Connection:** The connection between InfoMax and submodular/graphcut is that in Sec.3.3 when we explain why optimizing the quadratic optimization target defined in (2) is equivalent to finding the most informative subset, we use the Graph-cut-conditional gain (GCCG) instantiations in the sub-modular theory to proof this equivalence. This has been discussed in lines 1264-1266 in the revision.
>
>
> ---
>
> Q.2 The average time for InfoMax on each GPU is 4.7 hours, and the difference between the mean and the max value for InfoMax is only about 0.1 hours. This difference is generally caused by random factors, such as hardware cooling and voltage issues. For reference, the average time for D2-Pruning is 4.9 hours, while its max value is 5.7 hours, highlighting a more significant difference compared to the maximum running time of D2-Pruning. For this reason, D2-Pruning needs a greedy (non-parallel) process in data selection, and its serial process requires frequent interaction between CPU and GPU, which brings greater instability. Therefore, we can conclude that InfoMax has better performance, better stability, and faster speed than D2-Pruning, which has the best performance at present.
>
> About the overall total GPU hours: GPU hours measure the total running time on each GPU. For a lab or team capable of handling Billion-scale datasets, it will take less than 10 hours to complete effective pruning of 1B data using an additional 8 low-priced 3090 GPUs, or less than 5 hours using 16 of the same GPUs. This is very positive for the subsequent training (which is the most time-consuming process).
>
>
>
> Best regards,
>
> Submission1258 Authors

---

> ### Comment · Reviewer_2PGZ · 2024-11-26
>
> Thanks for the reply. But I think the statement further proves that the proposed method is only another formulation of graphcut, without major differences. The main contribution is located at the efficient solver, which provides more stable distributionally time consumption, but also only accelerates marginally over the previous method in terms of the total GPU hour. Considering the contribution and limitations, I will insist on rejecting the paper, but I will not further decrease the score.

---

> > ### Author Response · Authors · 2024-11-28
> > **Response to Reviewer 2PGZ**
> >
> > Thank you for your prompt response and your efforts in helping us refine our work! We would like to clarify the fundamental differences between Infomax and GraphCut:
> >
> > ---
> >
> > **Distinct Objectives:** InfoMax introduces a novel formulation for coreset selection as an information maximization problem. This approach represents a first for the coreset selection task, measuring the overall information of a selected coreset as a discrete quadratic function—a key contribution of our work. In contrast, GraphCut encompasses a family of techniques designed for modeling and solving combinatorial optimization problems. Since these methods address fundamentally different problem types, they are not directly comparable.
> >
> > **Optimization Approach:** InfoMax employs a continuous relaxation of the problem, allowing for the use of gradient-based solvers for efficient optimization. This approach is fundamentally different from some discrete GraphCut-based optimization strategies. The adoption of a gradient-based method aligns well with the continuous relaxation inherent in InfoMax’s formulation, while GraphCut’s discrete methods are not suitable for our framework. For example, applying the well-known Ford-Fulkerson graph-cut algorithm to process a sparse graph with over 100,000 edges (around 20,000 samples) can take tens of hours. In contrast, we implement a well-known efficient yet approximate graph cut method—Normalized Cut—as a comparison to our InfoMax-Solver. This method transforms the graph cut problem into a process for solving the smallest eigenvector of the graph Laplacian.
> >
> >
> > The specific results for the coreset on the CIFAR-10 dataset are presented in the table below. As we can see, the InfoMax-Solver consistently outperforms the GraphCut-based Solver across various selection ratio settings. This finding highlights the importance of tailoring algorithms to specific contexts. While general solvers like GraphCut may perform adequately in a broad range of scenarios, they often lack the effectiveness of algorithms specifically designed for particular tasks. Our results demonstrate that the specialized InfoMax-Solver can provide significant advantages in both performance and efficiency.
> >
> > The reason we did not conduct experiments on larger datasets (in the millions or billions) is that solving the Laplacian spectral decomposition for large-scale graphs is extremely time-consuming.
> >
> > Method|SR=10\%|SR=30\%|SR=70\%
> > ---|---|---|---
> > N-Cut|72.4|86.6|95.1
> > InfoMax-Solver|89.1|94.1|95.5
> >
> > We will include a detailed discussion of this distinction in the paper.
> >
> >
> > ---
> >
> >
> > **Concerns about efficiency.** We believe that the efficiency improvements offered by InfoMax are substantial. Specifically, InfoMax completes processing in 4.8 hours, compared to 5.7 hours for D2-Pruning—resulting in nearly a 20\% reduction in time. Moreover, InfoMax demonstrates the capability to handle large-scale datasets containing billions of data points using only 16 consumer-grade GPUs, completing the task in under 5 hours. In contrast, D2-Pruning takes 5.7 hours for the same task. This underscores InfoMax's status as both a highly efficient and cost-effective solution.
> >
> > Method on 1000M Samples|Graph Construction on 1000M Samples|cost in total on 1000M Samples| overall-time (2*8 GPUs) on 1000M Samples
> > ---|---|---|---
> > D2-Pruning|59 GPU-Hours |78.3 GPU-Hours | 5.7 Hours
> > InfoMax|59 GPU-Hours|75.2 GPU-Hours|4.8 Hours
> >
> >
> > Thank you once again for your response! We are pleased to engage in this discussion with you.
> >
> > Wishing you all the best!
> >
> > Authors of InfoMax

---

### Official Review · Reviewer_kwDs · 2024-11-03

**Soundness:** 2
**Presentation:** 2
**Contribution:** 2
**Rating:** 6
**Confidence:** 3

**Summary:**

A novel method, InfoMax, is proposed for data pruning and core set selection. The motivation of the method is finding a subset of samples that maximizes overall information by simultaneously considering each sample’s information contribution and the information overlap among them. The authors formulate the core set selection as a discrete quadratic programming (DQP) problem with equality constraints that specify the desired number of selected samples. And a robust gradient-based solver is proposed to address scalability. Extensive experiments conduct the best performance and consistently outperforms the state-of-the-art schemes in a series of tasks.

**Strengths:**

The proposed InfoMax designed to maximize overall information by accounting for each sample’s individual contribution while reducing information overlap, with a simultaneous focus on maintaining diversity and importance. And the proposed efficient gradient-based solver makes InfoMax scale to large-scale datasets. The proposed method brings performance enhancements in a series of different tasks.

**Weaknesses:**

Some typo: a) 95.7 is best result in CIFAR-10 70% setting from Tab. 1.  b) 51.8 should not be bolded in MMUL 10% setting from Tab. 3.
The ablation study for 4 hyper-parameters only conducted in multi-modality pre-training task. For classification and instruction tuning tasks, what kind of impact do the 4 parameters have on the final results?

**Questions:**

The selection of the four hyper-parameters depends on the specific dataset or task? If it depends on the specific dataset, how should the corresponding hyper-parameters be determined when faced with a new dataset?

---

> ### Author Response · Authors · 2024-11-23
> **Response to Reviewer kwDs**
>
> We greatly appreciate your constructive feedback on our work. If you have any additional concerns, please feel free to contact us. We are committed to addressing any issues you may raise. Should you find our response satisfactory, we would be thankful for a higher rating. Thank you for your consideration!
>
> ---
>
>
> **Weakness-1. The reviewer pointed out some typos.**
>
> We sincerely appreciate your careful review. We have corrected these typos in the revision!
>
>
>
>
> **Weakness-2. Hyper-parameter test on Classification and Instruction tuning.**
>
> We appreciate your helpful suggestion and have updated the revised paper accordingly; see Appendix B.3. Our ablation analysis examines key factors: the subset size after dataset partitioning, the value of $k$ in the K-NN graph, the pairwise weights $\alpha$ for the InfoMax targets, and the number of iterations $T$ for the InfoMax solver. We conducted experiments on classification tasks with ImageNet-22K (14 million samples) and SFT experiments for Llama-3-8B-Instruct using the OpenMathInstruct-v2 dataset (14 million math question-answer pairs). Details are found in Appendix B.3 of the revised paper.
>
> Here, we summarize the Table 8  in Appendix B.3 from the revision as follows.
>
>
> **As for the partition strategy**, we study the effect of each subset size on the final performance. When the size increases from 0.1M to 1M, the performance also increases by 3.85 top-1 acc for ImageNet-22K and 1.85 for OpenMathInstruct-v2. However, when the size increases from 1M to 2M, the performance improvements are 0.17 and 0.3 for ImageNet-22K and OpenMathInstruct-v2 respectively. The experimental result is consistent with the ablation for the partition strategy in Section 4.4 on CC12M, that is, when the subset size is greater than 1M, the performance improvement would be saturated. A larger subset size will yield better performance but will result in higher computational complexity. For a better trade-off between efficiency and performance, we set the partition strategy to ensure that each subset size is at least 1M.
>
> **Regarding the sparse rate $k$** (the size of the neighborhood when constructing the samples' k-NN graph), we also observed marginal performance improvements for both ImageNet-22K and OpenMathInstruct-v2 when $k \geq 5$ (e.g., increasing k from 5 to 200 only brings an improvement on performance by 0.06 for OpenMathInstruct-v2). Considering that larger values of $k$ often lead to increased computational complexity, we recommend maintaining $k = 5$ across different scenarios. This recommendation is consistent with the ablation study on the sparse rate $k$ presented in Section 4.4. This experiment demonstrates that InfoMax exhibits strong generalization capabilities for hyper-parameters across various scenarios.
>
> **Regarding the pairwise weight $\alpha$**, we found that its impact on performance generally follows a trend of initial improvement followed by a decline as $\alpha$ increases, consistent for both ImageNet-22K and OpenMathInstruct-v2. Notably, the optimal performance ranges for these datasets are between 0.01 to 10 and 0.3 to 3, respectively. Therefore, we recommend setting $\alpha = 0.3$. This recommendation aligns with the conclusions drawn from the ablation study in Section 4.4. This experiment illustrates that InfoMax demonstrates robust generalization capabilities for hyper-parameters across different scenarios.
>
> **Finally, for the number of iterations $T$**, increasing $T$ from 5 to 20 results in significant performance improvements of 1.78 and 2.72 for ImageNet-22K and OpenMathInstruct-v2, respectively. However, beyond this point, further increases yield only marginal benefits. For instance, increasing $T$ from 20 to 60 produces improvements of only 0.04 and 0.28 for ImageNet-22K and OpenMathInstruct-v2, respectively, while the computational complexity triples. Therefore, we recommend setting $T = 5$, which is consistent with the conclusions of the ablation study in Section 4.4. This further demonstrates the strong generalization capabilities of InfoMax regarding hyper-parameters across various scenarios.
>
>
> **In conclusion, our results match the conclusions in Section 4.4 about the vision-language pretraining task on CC12M for both experimental setups.**

---

### Official Review · Reviewer_tXt5 · 2024-11-04

**Soundness:** 2
**Presentation:** 2
**Contribution:** 2
**Rating:** 6
**Confidence:** 3

**Summary:**

The paper introduces InfoMax, a novel data pruning method aimed at maximizing information content while minimizing redundancy in selected samples. It measures sample information through importance scores and quantifies redundancy using pairwise sample similarities. The coreset selection problem is formalized as a discrete quadratic programming task. An efficient gradient-based solver is proposed, along with sparsification techniques and dataset partitioning strategies, to scale InfoMax to large datasets. The significance lies in its ability to enhance model training efficiency and data storage without compromising performance.

**Strengths:**

1.) The paper is well-organized, with clear explanations of complex concepts, making it accessible to a broad readers.
2.) InfoMax offers a new perspective on data pruning by focusing on information maximization, formalizing the problem as a quadratic programming task and offering a clear explanation of the underlying information theory.
3.) Extensive experiments across diverse datasets and tasks validate the effectiveness of InfoMax, showing consistent improvements over existing methods.

**Weaknesses:**

1.) The method's reliance on calculating pairwise similarities and the construction of the similarity matrix may become computationally intensive and could be further optimized.
2.) The performance of InfoMax is sensitive to hyperparameters like the partition rate, sparse rate, and pairwise weight, which may require careful tuning for different datasets.
3.) The paper could provide more analysis on the risk of overfitting when using the gradient-based solver, especially with high pruning ratios.

**Questions:**

see the Weaknesses

---

> ### Author Response · Authors · 2024-11-23
> **Response to Reviewer tXt5 (Part-1)**
>
> We sincerely appreciate your constructive feedback on our work. Please do not hesitate to reach out if you have any further concerns. We are eager to address any issues you may have. Thank you!
>
> ---
>
> **Weakness-1. The method's reliance on calculating pairwise similarities and the construction of the similarity matrix may become computationally intensive and could be further optimized.**
>
> Thanks!
>
> Measuring pairwise similarities is key to improving diversity and reducing redundancy in optimization. Many methods, like K-center and D2-Pruning, rely on this step.
>
> Calculating similarity for all data pairs takes a lot of time, with a complexity of $O(N^2)$, where N is the number of samples. Instead, we can create a K-NN graph by focusing only on the nearest K samples for each data point, which reduces the complexity to $O(Nk)$. Using tools like FAISS makes this easier. For example, FAISS [1] can build a K-NN graph for 12 million multi-modal data points within an hour.
>
>
>
> **Weakness-2. Sensitivity test of some hyper-parameters, partition rate, sparse rate, and pairwise weight, which may require careful tuning for different datasets.**
>
>
> We appreciate your helpful suggestion and have updated the revised paper accordingly; see Appendix B.3. Our ablation analysis examines key factors: the subset size after dataset partitioning, the value of $k$ in the K-NN graph, the pairwise weights $\alpha$ for the InfoMax targets, and the number of iterations $T$ for the InfoMax solver.
> We conducted experiments on classification tasks with ImageNet-22K (14 million samples) and SFT experiments for Llama-3-8B-Instruct using the OpenMathInstruct-v2 dataset (14 million math question-answer pairs). Details are found in Appendix B.3 of the revised paper.
>
> Here, we summarize the Table 8  in Appendix B.3 from the revision as follows.
>
>
> **As for the partition strategy**, we study the effect of each subset size on the final performance. When the size increases from 0.1M to 1M, the performance also increases by 3.85 top-1 acc for ImageNet-22K and 1.85 for OpenMathInstruct-v2. However, when the size increases from 1M to 2M, the performance improvements are 0.17 and 0.3 for ImageNet-22K and OpenMathInstruct-v2 respectively. The experimental result is consistent with the ablation for the partition strategy in Section 4.4 on CC12M, that is, when the subset size is greater than 1M, the performance improvement would be saturated. A larger subset size will yield better performance but will result in higher computational complexity. For a better trade-off between efficiency and performance, we set the partition strategy to ensure that each subset size is at least 1M.
>
> **Regarding the sparse rate $k$** (the size of the neighborhood when constructing the samples' k-NN graph), we also observed marginal performance improvements for both ImageNet-22K and OpenMathInstruct-v2 when $k \geq 5$ (e.g., increasing k from 5 to 200 only brings an improvement on performance by 0.06 for OpenMathInstruct-v2). Considering that larger values of $k$ often lead to increased computational complexity, we recommend maintaining $k = 5$ across different scenarios. This recommendation is consistent with the ablation study on the sparse rate $k$ presented in Section 4.4. This experiment demonstrates that InfoMax exhibits strong generalization capabilities for hyper-parameters across various scenarios.
>
> **Regarding the pairwise weight $\alpha$**, we found that its impact on performance generally follows a trend of initial improvement followed by a decline as $\alpha$ increases, consistent for both ImageNet-22K and OpenMathInstruct-v2. Notably, the optimal performance ranges for these datasets are between 0.01 to 10 and 0.3 to 3, respectively. Therefore, we recommend setting $\alpha = 0.3$. This recommendation aligns with the conclusions drawn from the ablation study in Section 4.4. This experiment illustrates that InfoMax demonstrates robust generalization capabilities for hyper-parameters across different scenarios.
>
> **Finally, for the number of iterations $T$**, increasing $T$ from 5 to 20 results in significant performance improvements of 1.78 and 2.72 for ImageNet-22K and OpenMathInstruct-v2, respectively. However, beyond this point, further increases yield only marginal benefits. For instance, increasing $T$ from 20 to 60 produces improvements of only 0.04 and 0.28 for ImageNet-22K and OpenMathInstruct-v2, respectively, while the computational complexity triples. Therefore, we recommend setting $T = 5$, which is consistent with the conclusions of the ablation study in Section 4.4. This further demonstrates the strong generalization capabilities of InfoMax regarding hyper-parameters across various scenarios.
>
>
> **In conclusion, our results match the conclusions in Section 4.4 about the vision-language pretraining task on CC12M for both experimental setups.**

---

> > ### Comment · Reviewer_tXt5 · 2024-11-28
> >
> > Thank you for your rebuttal, it has resolved most of my doubts.

---

> > > ### Author Response · Authors · 2024-11-28
> > > **Response to Reviewer tXt5**
> > >
> > > We are sincerely appreciated for your reply! If you have any further questions please feel free to reach out to us. We are more than willing to address your concerns.
> > >
> > > Best regards
> > >
> > > Authors of InfoMax

---

> ### Author Response · Authors · 2024-11-24
> **Response to Reviewer tXt5 (Part-2)**
>
> **Weakness-3. The paper could provide more analysis of the risk of overfitting when using the gradient-based solver, especially with high pruning ratios. (show results under the setting of PR less than 1\%).**
>
> Thanks!
>
> InfoMax is not a machine learning model; it’s a data processing algorithm designed to find the best coreset with the most information, as described in Eq.(2) of the main paper. It uses a gradient-based method to solve this optimization problem, so it doesn’t overfit like a machine learning model.
>
> We compare InfoMax with D2-Pruning and CCS at an extremely high pruning ratio, and InfoMax still shows strong performance.
>
> Pruning ratio|IN-1K 99\%|IN-1K 99.5\%|IN-1K 99.9\%
> ---|---|---|---
> CCS|9.8|6.4|0.9
> D2-Pruning|7.5|7.3|1.2
> InfoMax|11.0|8.9|2.1
>
>
>
> [1]. FAISS: Billion-scale similarity search with GPUs. IEEE Transactions on Big Data.
>
> [2]. D2 Pruning: Message Passing for Balancing Diversity and Difficulty in Data Pruning. ICLR 2024.
>
> [3]. Coverage-centric Coreset Selection for High Pruning Rates. ICLR 2023.

---

### Official Review · Reviewer_HVmb · 2024-11-04

**Soundness:** 3
**Presentation:** 2
**Contribution:** 3
**Rating:** 3
**Confidence:** 4

**Summary:**

The article introduces a novel data pruning method called InfoMax, which is also known as coreset selection, designed to maximize the information content of selected samples while minimizing redundancy. The proposal of the InfoMax algorithm maximizes overall information by considering both individual contributions and information overlap of samples. The development of an efficient gradient-based solver is enhanced by sparsification techniques and dataset partitioning strategies, enabling InfoMax to scale to large-scale datasets. Extensive experiments demonstrating InfoMax's superior performance across various data pruning tasks, including image classification, vision-language pre-training, and instruction tuning for large language models.

**Strengths:**

- InfoMax can effectively handle datasets with millions of samples within tens of minutes through sparsification techniques and dataset partitioning strategies.
- InfoMax shows better performance compared to existing methods, especially under high pruning ratios.
- InfoMax exhibits strong generalization capabilities across different datasets and tasks, including cross-model and cross-setting generalization.

**Weaknesses:**

- The Introduction in this paper lacks a high-level insight of the InfoMax to explain why it could work better than \(D^2\) Pruning, which makes the reader difficult to understand the motivation of the proposed pruning algorithm intuitively, for example, what is the more intuitive motivation of the proposed work to maintain a proper balance between importance and diversity?
- In line#243, K_{z,s} should be inter-sample redundancy instead of intra-sample redundancy.
- The symbolic sign used in the method lacks clarity and conciseness. For example, I and X are repeated frequently with different and dazzling superscripts and subscripts.
- The length of the main text disobeyed the strict limits of 10 pages, because in the current typesetting the Conclusion section belongs to the main text.
- In my opinion, the contribution and novelty of this work is limited due to its start points similar to \(D^2\) Pruning, except the only scalable solver (step 4 in Figure 4).

**Questions:**

- What is the set-level information of the candidate subset of \(D^2\) Pruning? What is its difference from InfoMax?
- Why can \(D^2\) Pruning perform better than other hybrid approaches from a more intuitive perspective? I understand the content of Section3.3 but I hope the authors could convert the Section 3.3 into a more high-level and insightful motivation, which could be placed in Section 1 for more readers to get the key insight.

---

> ### Author Response · Authors · 2024-11-23
> **Response to Reviewer HVmb (Weakness part)**
>
> We sincerely appreciate your constructive comments on our work! If there is any additional concern, please let us know! We are glad to solve your concerns! If you are satisfied with our response, we hope to get a higher rating!
>
> ---
>
> **Weakness-1. lacks a high-level insight to explain why it could work better than D2-Pruning.**
>
>
> This is an insightful question! We also highly recommend the reviewer see Appendix.F in the revision for the discussion: why InfoMax outperforms other works like D2-Pruning.
>
>
> (a). D2-Pruning Overview. D2-Pruning is a local optimization method inspired by graph message passing. It uses a greedy strategy, where datasets are represented as graphs. Each node represents a sample, with its value indicating informativeness, and edges show similarities between samples. The algorithm iteratively selects the most informative nodes but may reduce the scores of similar nodes to minimize redundancy. However, its greedy nature can lead to suboptimal solutions, making it hard to balance importance and diversity.
>
> (b). InfoMax Approach. In contrast, InfoMax takes a global approach to data pruning by maximizing the informativeness of samples while minimizing redundancy. This method aims to identify the most informative subset of data rather than getting stuck in local solutions. InfoMax uses an efficient solver based on proximal gradient techniques, ensuring consistent convergence and better overall results.
>
> (c). Experimental Analysis. To compare the two methods, we conducted experiments on ImageNet with a selection ratio of 10\%. We analyzed key metrics: mean redundancy (average similarity among samples) and mean informativeness (average score value per sample). The coreset generated by InfoMax showed higher informativeness and lower redundancy, leading to improved performance in the model trained on this coreset.
>
>
> Method|Mean-informativeness ($\uparrow$)|Mean-redundancy ($\downarrow$)|Top-1 Accuracy ($\uparrow$)
> ---|---|---|---
> D2-Pruning|0.491|0.292|55.6
> InfoMax|0.563|0.216|59.0
>
>
>
>
>
> **Weakness-2. The reviewer pointed out some typos.**
>
> Thanks for your constructive suggestion! We have corrected them in the revision.
>
>
>
>
> **Weakness-3. The reviewer thinks the symbolic sign lacks clarity.**
>
> Thank you! We have added some new symbolic conventions in Table 4 (highlighted in blue). If you have any questions or find any symbols unclear, please let us know—we are happy to address your concerns!
>
>
> **Weakness-4. About the paper length.**
>
> Thanks! We have moved the Limitation section to the Appendix to improve the typography.
>
>
>
> **Weakness-5. Concerns on novelty comparison with D2-Pruning.**
>
> InfoMax and D2-Pruning differ significantly in terms of their motivation, solution methods, and performance. We also highly recommend the reviewer see Appendix.F in the revision for the discussion: why InfoMax outperforms other works like D2-Pruning.
>
> Our main innovation is turning the data pruning problem into a unified combinatorial optimization focused on maximizing information. Under certain conditions described in Section 3.3, we simplify a complex information maximization problem into a more manageable second-order combinatorial problem. This approach aims to maximize the importance of individual samples while minimizing redundancy between samples. We also created an efficient solver for this problem, and our method, InfoMax, performs better than others in various scenarios.
>
> In contrast, D2-Pruning uses a different approach based on graph message passing. In this method, each sample's score is treated as a node value, and the similarity between samples is treated as an edge value. D2-Pruning uses a greedy selection strategy, picking the highest-scoring samples and reducing the scores of neighboring samples to avoid redundancy. However, this can lead to suboptimal results. In contrast, InfoMax optimizes a global view of information, effectively integrating sample diversity and information more thoroughly.

---

> ### Author Response · Authors · 2024-11-23
> **Response to Reviewer HVmb (Question part)**
>
> **Question-1. What is the set-level information of the candidate subset of D2-Pruning? What is its difference from InfoMax?**
>
> Thank you!
>
> The set-level information measures the total information content of a sample set, as outlined in Eq.(4). In this paper, we reformulate the coreset selection or data pruning as seeking the subset with the maximum information. We need to clarify that this concept is a key defined in our work (firstly).
>
> Note that the D2-Pruning paper does not provide any related definitions and discussion, as it is inspired by a message-passing mechanism on graphs. But, we have also analyzed the D2-Pruning from the information framework of why it performs sub-optimally, see Appendix.F in the revision for details. D2-Pruning operates greedily, selecting samples that are least similar to those already chosen (minimizing mutual information) while also maximizing their score (intra-sample information). However, the greedy nature of D2-Pruning often results in sub-optimal performance.
> In contrast, our InfoMax directly aims to optimize for maximum set-level information, as defined in Eq. (4), to find the globally best coreset.
>
>
>
>
>
> **Question 2. Why can D2Pruning perform better than other hybrid approaches from a more intuitive perspective?**
>
> That's a great question! We also discussed why D2-Pruning outperforms other previous hybrid methods in detail, see Appendix.F in the revision for details.
>
> Before D2Pruning, many hybrid data pruning methods divided data into groups, either by clustering similar features [2] or by splitting score distributions evenly [1]. From these groups, samples were either picked at random [1] or based on their highest scores [2].
>
> However, these methods often fail to provide diverse samples within each group, leading to poorer performance compared to D2Pruning. D2Pruning improves this process by evaluating both how redundant a sample is compared to those already chosen and how important each sample is at each step of selection.
>
>
> [1]. Coverage-centric Coreset Selection for High Pruning Rates. ICLR 2023.
>
> [2] Dos: Diverse outlier sampling for out-of-distribution detection. ICLR 2024.

---

> ### Author Response · Authors · 2024-11-25
> **Looking forward to your further reply**
>
> Dear Reviewer HVmb:
>
> We sincerely thank you for your efforts in reviewing our paper and your suggestions for enhancing this work. As we are approaching the end of the discussion period, we would like to ask whether there are any remaining concerns regarding our paper or our response. We are happy to answer any further questions.
>
> Best regards,
>
> Authors of InfoMax

---

> > ### Comment · Reviewer_HVmb · 2024-11-25
> >
> > Thank you for your response. However, according to the submission policy (https://iclr.cc/Conferences/2025/CallForPapers), the page limit applies to both the initial and final camera ready version. I would maintain my initial rating to guarantee justice for all the other submissions.

---

> > > ### Author Response · Authors · 2024-11-25
> > > **Response to Reviewer HVmb**
> > >
> > > Thank you for your feedback!
> > >
> > > We would like to clarify that **the portion of our submission exceeding the length limit pertains to an additional Limitation Statement, but not the main body of the paper**. And it occupies only one and a half lines. This was a minor formatting oversight made before the submission deadline. **We believe this does not create any unfair advantage, as there is plenty of free space in the text, and we can easily compress the whole thing into less than 10 pages**.
> > >
> > > Best regards,
> > >
> > > Submission1258 Authors

---

> > > > ### Comment · Reviewer_HVmb · 2024-11-25
> > > >
> > > > I understand your point. But the oversight is unacceptable from my perspective. I double-checked that the limitation part is included in the Conclusion Section, which obviously disobeyed the submission policy. On this point, I hold on to my principle that the work should be desk-rejected. I will leave the SACs and ACs to make their final decision as this review is open to the public.

---

> > > > ### Comment · Reviewer_2PGZ · 2024-11-25
> > > >
> > > > According to the author guide (https://iclr.cc/Conferences/2025/AuthorGuide), the ethics statement and reproducibility statement can have extra space, but not limitation statement.

---

> ### Author Response · Authors · 2024-11-25
> **Response to Reviewer HVmb**
>
> Dear Reviewer,
>
> We sincerely appreciate your time and feedback on our work.
>
> During the rebuttal period, we invested considerable effort to address the concerns raised by you and the other reviewers. We sincerely hope to receive your evaluation based on the content and quality of our paper, as your insights are invaluable for helping us improve our work. We are eager to make a positive contribution to the data-centric academic community.
>
> Additionally, we would like to clarify that the issue regarding the manuscript's length pertains solely to the discussion of limitations and does not confer any substantive advantage regarding the acceptance of the paper.
>
> All authors of InfoMax have invested considerable time and effort into this work, and we kindly ask for your unbiased assessment to be based on its actual content. If you have any further questions or concerns, we warmly encourage you to share them with the AC and SAC.
>
> Thank you once again for your thoughtful review. We sincerely appreciate it.
>
> Best regards,
>
> Submission1258 Authors

---

> > ### Comment · Reviewer_HVmb · 2024-11-25
> >
> > Sure. That is why I still give a careful review of your work, and I hope to help you better improve your work. Your revision is responsible for my suggestions and I appreciate your effort in your work. But with the submission policy issue, I have my own principle to stick to the "call for paper" requirement. Thank you for your understanding. Good luck.

---

> > > ### Author Response · Authors · 2024-11-25
> > > **Response to Reviewer HVmb**
> > >
> > > Dear Reviewer HVmb,
> > >
> > > Thank you for taking the time to review our work and for your valuable feedback. We are very pleased to hear that you are satisfied with our responses during the rebuttal phase.
> > >
> > > We would like to further understand your views on the content of our paper. Based on our current replies and the content of the paper, do you have any additional questions or suggestions regarding the content, experiments, or methods? Additionally, we would sincerely appreciate your rating of the paper based on the content and methodology. We greatly value your assessment.
> > >
> > > Thank you once again for your time and effort. We look forward to your response.
> > >
> > > Wish you all the best,
> > >
> > > Submission1258 Authors

---

> > > ### Author Response · Authors · 2024-11-25
> > > **Response to "Official Comment by Reviewer HVmb"**
> > >
> > > Thanks for your reply and your time!
> > >
> > > 1. Note that the Limitation statement does not belong to the conclusion section. The conclusion is the end of the main body (main text) of our paper, as it summarizes the content of this paper.  However, the Limitation statement is additional information that is not shown in the main body of the paper. It is just an independent statement. Hence, it doesn't belong to the main text.
> > >
> > > 2. On the other hand, if we want to include the Limitation Statement in the main text, we **will not** make it an independent part as now, but just continue it directly after the conclusion paragraph text, which is also very smooth to read.
> > >
> > > 3. Note that the "call for paper" requirement strictly requires the <MAIN TEXT> no more than ten pages. Hence, the Limitation statement should not be rejected by the desk because it is an independent part that is just behind the Conclusion section (the end of the main text).
> > >
> > >
> > > **During the rebuttal period, we invested considerable effort to address the concerns raised by you and the other reviewers. We sincerely hope to receive your evaluation based on the content and quality of our paper, as your insights are valuable for helping us improve our work. We are eager to make a positive contribution to the data-centric academic community. We kindly ask for your unbiased assessment to be based on its actual content.**
> > >
> > > Best regards,
> > >
> > > Submission1258 Authors

---

> ### Author Response · Authors · 2024-11-25
> **Response to Reviewer 2PGZ**
>
> Thank you for your discussion!
>
> 1. We would like to clarify that the portion of our submission exceeding the length limit (by only 1.5 lines) belongs to the Limitation Statement part, not the main body of the paper. It does not belong to the conclusion section, as its content is not summarized from the main text. **It is just an independent statement.** The page limit applies only to the main text, as stated in the [Call for Papers](https://iclr.cc/Conferences/2025/CallForPapers).
>
> 2. We included this statement after the conclusion to inform readers of the limitations of our work. It is important to note that this does not contribute positively to the paper's rating. Therefore, we believe it does not create any unfairness within the academic community.
>
> 3. Additionally, this section occupies only one and a half lines. This was a minor formatting oversight before the submission deadline. We believe it does not confer any unfairness, as there is ample space in the text, and we could easily condense the entire submission to fit within the 10-page limit, as demonstrated in our most recent revision of InfoMax.
>
> 4. We would sincerely appreciate your rating of the paper based on the content and methodology. We greatly value your assessment.
>
> Best regards,
> Submission1258 Authors
>
> ---

---

> > ### Comment · Reviewer_HVmb · 2024-11-26
> > **Final response**
> >
> > I agree with Reviewer 2PGZ.
> > Again, following both the call for paper and the author guide, there is no any statement claiming that the limitation part could be over 10 pages. Besides, I confirmed that the limitation part did not be separated as an independent section. As I said, the review would be open to the public, and ACs would make their own decisions.
> > We reviewers have our own judgment on the content and rules. Please note that that's the overall rating including consideration of the paper format and submission policy. Please respect our decision. Thank you again.

---

> ### Author Response · Authors · 2024-11-29
> **Response to Reviewer HVmb**
>
> Thank you for your time and effort dedicated to reviewing our paper. We truly appreciate your valuable input and are more than happy to address any concerns you may have.
>
> Best regards,
>
> Submission1258 Authors

---

### Official Review · Reviewer_nGro · 2024-11-04

**Soundness:** 3
**Presentation:** 4
**Contribution:** 3
**Rating:** 8
**Confidence:** 3

**Summary:**

This work presents a novel approach to data pruning, that aims to maximise the core set information through the simultaneous optimisation of individual sample information and overlapping information between samples. To solve this equality the authors present this as a discrete quadratic programming problem, presenting an efficient gradient based solver to improve the scalability of the method. The resulting core set’s are tested in a variety of settings to demonstrate performance and information preservation, where the authors demonstrate significant improvements over prior methods. Furthermore, the authors demonstrate their findings across a variety of tasks, and datasets to present generalisation of the method. A full sensitivity analysis is presented alongside all key details for reproducibility, providing readers with the necessary information to apply or replicate the findings.

**Strengths:**

**Structure and Clarity:**
- The work is well organised and presented clearly defining the problem statement hypothesis of the work. The core narrative and all technical contributions are written in a clear and concise manner, guiding most readers well to fully understand the contributions.
- Most of the key concepts discussed are presented in the form of visualisations, or figures which help justify the narrative, and provide evidential basis of the investigations.

**Method, hypothesis, findings, and rationale:**
- The method is well justified with Figure 2 presenting clear empirical evidence to claims of prior method weaknesses and properties. This figure alone does a lot of the heavy lifting in providing strong rationale behind the decision making which defines the proposed method.
- Method description itself is sensible, clearly presented and interpretable. The method section is further supported with proofs and significant empirical findings justifying decisions made.
- The findings demonstrate the proposed method is highly performant when compared to the selected benchmark methods and on the selected tasks.
- Broader impact statement is provided and some limitations are addressed.

**Reproducibility:**
- All details are presented for full reproduction in the text including optimisation, hyperparameterisation, datasets, and architectural settings. The algorithm presented in the appendix is a nice addition to support reproducibility, and the
- Descriptions of empirical setups are present and clear to follow,

**Experimental results:**
- Generally, the authors do a good job covering a variety of empirical evaluations to test their method, presenting results on a variety of datasets for different tasks.
- The sensitivity analysis is a nice addition, presenting the trade-off between the first and second order terms. While also having a secondary effect of demonstrating the robustness of the method to hyperparamter changes.
- While additional experimentation would improve the work (see weaknesses) the core empirical evaluation does a good job of conveying the core message and rationale of the work

**Weaknesses:**

**Empirical Comparisons**
- How does the method perform when compared to other data pruning methods not included in this work such as Sieve And Dyn-Unc. If there is a strong reason for not including these works then please correct me on this point.
- The results in table 1 could be considered misleading with incorrect bolding of top results. For 70% cifar10 d2 is performing better, yet, infomax is highlighted. I assume this is a simple mistake.
- Computational compression between methods is performed but only at a small scale, against d2 and entropy. The addition to all analysed works would further support the statement that InfoMax is computationally efficient while being performant. It is hard from this small scale evaluation the trade-off between these points.
- The addition of a more explicit ablation would be nice to have. How does changing the instantiation of the first order term I(z) effect the performance for example?

**Data Generalisation of method:**
- The authors method employs pre-trained networks to produce representations on which the method is applied (shown in Figure 3). However, from my understanding, the learnt representations themselves are produced from learning . This leads to two distinct weaknesses.
- The first is that the authors do not evaluate the performance and generalisation of the method to produce core sets when applying a different dataset. Thus, how does the method perform if the core feature extractor itself has not been trained on the data in question.
- Secondly, this leads to a distinct contradiction of the work if it is require to re-train this network per dataset, thus meaning full datasets will have to be trained on before pruning can occur. Thus, leading to the desired computational cost decrease being lost.
- Furthermore, the choice of SSL method could be highly impactful here, it has been shown that some methods akin to MSN and DINO are both optimising to preserve maximal information across samples than say predictive methods such as BYOL. Therefore, have the authors considered how different pertained models to produce features may interact with the criteria for pruning proposed?

**Further analysis on the selected core set:**
- Does the coreset exhibit the properties you are aiming to persevere? While the empirical evaluation provide strong evidence that the method does produce a notion that the coreset is indeed informative, are there any other analyses that could be performed to provide further evidence that information redundancy is improved?
- While I am happy to be corrected on the above point, it does feel that some simpler and more explicit analyses could provide interesting introspective of the selected coreset prompting insights for future works.

**Minor:**
- Paragraph starting at line 81 could be moved to preliminaries, its place in the introduction does not flow naturally.
- Error’s are not reported on empirical results, while you state that the SD lies within 0.85, this is considerable in some comparisons, therefore these should be given.
- The paper is marginally over the page limit, but this is likely due to a formatting error which can easily be addressed.

**Questions:**

1. Have the authors considered testing the method under noisy data distributions? If not, why not?
2. Where do you see the future of this work? You mention larger scale experimentation, but from a methodological perspective where are the obvious gaps / limitations?
- Most questions are posed in the weaknesses section to improve clarity of the question.

---

> ### Author Response · Authors · 2024-11-23
> **Response to Reviewer nGro (Part-1)**
>
> We sincerely appreciate your recognition and constructive comments on our work! If there is any additional concern, please let us know! We are glad to solve your concerns! If you are satisfied with our response, we hope to get a higher rating!
>
> ---
>
> **Weakness-1.1 Discussion and comparison with two new works.**
>
>
> Thank you for sharing these two references. Both works use score-based methods for large datasets and can work well with InfoMax as a sample-wise scoring methods. We've added them to our related works section.
>
> SIEVE [1] offers a stronger scoring method to evaluate vision-language datasets, serving as an alternative to the popular CLIP score. Dyn-Unc [2] introduces an uncertainty-based score that considers training dynamics, primarily for image classification datasets.
>
> In our experiments, we tested on CC12M (SR=10\%) to compare SIEVE with InfoMax, and on ImageNet-1K (SR=10\%) to compare Dyn-Unc with InfoMax. The results show that InfoMax significantly improves performance over these methods.
>
> Method|CC12M (Linear Prob on ImageNet-1K)|ImageNet-1K
> ---|---|---
> Dyn-Unc|-|14.4
> InfoMax + Dyn-Unc|-|58.2 (+43.8)
> SIEVE|48.7|-
> InfoMax + SIEVE|52.2 (+3.5)|-
>
> [1]. Sieve: Multimodal Dataset Pruning Using Image Captioning Models. CVPR 2024.
> [2]. Dyn-Unc: Large-scale Dataset Pruning with Dynamic Uncertainty. CVPR Workshop 2024.
>
>
>
> **Weakness-1.2 Mis-bolded value.**
>
> Thanks very much!! We have corrected this typo!
>
> **Weakness-1.3. Further speed comparison on large scale dataset.**
>
> Thank you for the great question! We compare the speed of InfoMax and D2-Pruning with larger datasets of 100 million and 1 billion samples. We measure performance in GPU hours.
>
> For both methods, we use the same strategy by randomly splitting the dataset into subsets of 1 million samples. The K-NN graph is built using FAiSS with k set to 5. For both InfoMax and D2-Pruning, which need graph construction, we used the Efficiency Enhancement Techniques from Section 3.2. Overall, InfoMax is faster than D2-Pruning because it can run its process in parallel on a GPU, while D2-Pruning's greedy selection has to be done one step at a time.
>
> Method|100M|1000M
> ---|---|---
> D2-Pruning|8.7|78.3
> InfoMax|7.4|75.2
>
>
>
>
> **Weakness-1.3. Additional explicit ablation on changing the instantiation of the first-order term I(z).**
>
> Great question! We recommend checking Table 6 in the Appendix, where we show different options for I(z), like the Forgetting score and Margin score, and also use different features for the kernel term, such as VQGAN features. InfoMax performs well in all these scenarios.
>
> Additionally, we introduce new options for I(z) in response to Weakness-1.1, including SIEVE and Dyn-Unc. InfoMax consistently delivers strong performance with all these different choices.
>
>
> **Weakness-2: Concerns about the data/feature generalization of the method.**
>
> Good question!
>
> In this paper, we discuss the choice of feature extractor. It doesn't need to be the same network used for the target dataset.
>
> We suggest the reviewer look at the results for InfoMax (unsupervised) in Table 1, where DINO-V2, a popular self-supervised feature extractor, is used. The informativeness measurement for InfoMax is the SSP score, which assesses how far a sample is from its cluster center in the DINO feature space. InfoMax (unsupervised) shows very strong results.
>
> In unsupervised settings, InfoMax consistently performs better than other methods. Even with a high pruning ratio, it can match or surpass many supervised approaches. For instance, at a 10\% selection rate, InfoMax on ImageNet outperforms the supervised D2-Pruning by 1.2\%.
>
> This demonstrates that InfoMax is highly effective across various conditions. In Table 6, we also test InfoMax using another unsupervised feature extractor, VQGAN.
>
> Additionally, we evaluate InfoMax with BYOL and MSN features on ImageNet-1K, again using a 10\% selection rate and measuring informativeness with the SSP score.
>
> Feature extractor|Top-1 Acc on ImageNet-1K
> ---|---
> EL2N|12.9
> D2-Pruning|55.6
> InfoMax|59.0
> InfoMax with BYOL|50.9
> InfoMax with MSN|54.2
>
> BYOL: Bootstrap your own latent: A new approach to self-supervised Learning. NeurIPS 2020.
>
> MSN: Masked Siamese Networks for Label-Efficient Learning. ECCV 2022.

---

> ### Author Response · Authors · 2024-11-23
> **Response to Reviewer nGro (Part-2)**
>
> **Weakness-3: Further analysis of the selected coreset.**
>
> That's a great suggestion! Thank you!
>
> We ran experiments with a 10\% selection ratio on ImageNet. We analyzed InfoMax, D2-Pruning, El2N, and K-center based on coreset measurements like mean redundancy and mean informativeness.
>
> Mean redundancy measures how similar the samples are on average, while mean informativeness measures the average score for each sample. We found that El2N had very high redundancy, and K-center had low mean informativeness, resulting in poorer performance for both methods.
>
> In contrast, the coreset selected by InfoMax showed higher informativeness and lower redundancy than D2-Pruning, leading to better performance for the model trained on that coreset.
>
> Method|Mean-informativeness ($\uparrow$)|Mean-redundancy ($\downarrow$)|Top-1 Accuracy ($\uparrow$)
> ---|---|---|---
> EL2N|0.726|0.743|12.9
> K-center|0.179|0.130|42.0
> D2-Pruning|0.491|0.292|55.6
> InfoMax|0.563|0.216|59.0
>
> **Minor weakness terms.**
>
> Great suggestion! We’ve revised the paragraph starting at line 81 for better flow. We've also included the standard deviation (STD) values for our methods in Appendix E.
>
> ---
>
> **Question-1: Have the authors considered testing the method under noisy data distributions?**
>
> Good question! The experiments were conducted on CC12M with noisy data settings. The original CC12M contains numerous incorrect samples, and this type of noise closely resembles the data noise present in real-world application scenarios.
>
> **Question-2: Where do you see the future of this work also with the challenges?**
>
> Great question! We think future work should focus on two main areas:
>
> (a). Application Scenarios: We want to test how well InfoMax works in different situations, like ImageNet/Video Generation and LLM pretraining. The main challenge is handling the huge amounts of data. Since InfoMax and D2Pruning need to create graphs, it's not feasible to do this for the entire dataset. However, we found that splitting a large dataset into smaller parts and then selecting coresets from each part can greatly improve efficiency and results.
>
> (b). Improving Method Designs: Right now, Unsupervised InfoMax performs a bit worse when using general unsupervised feature extractors compared to those trained specifically on the target dataset. We need to investigate why this happens and find ways to enhance performance. This would mean we won't need an extra trained feature extractor for coreset selection in the future.

---

> > ### Comment · Reviewer_nGro · 2024-11-24
> > **Response to rebuttal**
> >
> > Thank you to the authors for their highly informative and friendly rebuttal.
> >
> > I appreciate the clarification on areas of misunderstanding and results that I had missed from the appendix, my apologies for this.
> >
> > All my requests for additional evaluation had been provided and added to the revised text. I would stress that the authors add some further cross references and short descriptions to the main text to highlight these supplementary results.
> >
> > Overall, all of my questions have been addressed, new results provided and the manuscript revised. Therefore I have increased my score.

---

> > > ### Author Response · Authors · 2024-11-24
> > > **Thanks for the response from Reviewer nGro**
> > >
> > > Thank you for your response!
> > >
> > > We are pleased to receive your recognition! We have included the experimental results from the large-scale tasks in the Appendix and will incorporate cross-references and brief descriptions in the main text, as you suggested.
> > >
> > > Thank you again for your feedback!

---

### Official Review · Reviewer_J8Fq · 2024-11-06

**Soundness:** 2
**Presentation:** 3
**Contribution:** 3
**Rating:** 8
**Confidence:** 3

**Summary:**

This paper introduces InfoMax, a novel data pruning method designed to maximize the information content of selected samples while minimizing overlap. The authors formulate this objective as a discrete quadratic programming problem, which they then relax and solve using an efficient gradient-based approach. Experimental results demonstrate the substantial effectiveness of InfoMax, underscoring its potential in data-centric applications such as image classification, multi-modal pretraining, and instruction tuning for large language models.

**Strengths:**

1. The paper presents an elegant and well-formulated approach to data pruning, with a solid theoretical foundation that supports its design.
2. The authors conduct a diverse set of experiments, including pretraining vision-language and fine-tuning LLM, further strengthening the validation of the method.
3. The performance of InfoMax is impressive, achieving high accuracy in various applications and outperforming existing state-of-the-art methods in many cases.

**Weaknesses:**

1. Some notations in the paper are unclear. For example, the symbols \( P \) on line 163 and \( z_n \) on line 1104 lack sufficient explanation. Furthermore, the variable \( X_t \) in lines 1054 to 1067 should be bolded for consistency.
2. The motivation behind InfoMax is not entirely novel, as the concepts of diversity and importance (information) have been previously discussed in [1, 2].
3. The paper does not include comparisons with some relevant baseline methods, such as geometry-based methods.

[1] "D2 pruning: Message passing for balancing diversity & difficulty in data pruning", ICLR 2024.
[2] "Dos: Diverse outlier sampling for out-of-distribution detection," ICLR 2024.

**Questions:**

The authors argue that $D^2$-Pruning may result in suboptimal solutions due to its greedy selection process. However, the relaxation applied to the quadratic optimization problem in Eq. 7 is not proven to produce solutions consistent with the original formulation, which could also result in suboptimal solutions. Given this, what factors contribute to the improved performance of your method compared to $D^2$-Pruning?

---

> ### Author Response · Authors · 2024-11-23
> **Response to Reviewer J8Fq**
>
> Thank you for reviewing our articles and for your feedback! We will work to address your concerns in our replies. We look forward to discussing this further and hope to earn a higher rating from you!
>
>
> **Weakness-1: Some notations in the paper are unclear.**
>
> Thanks for your careful review! We have added further clarifications and highlighted them in blue!
>
> **Weakness-2: Concerns about the motivation behind InfoMax given [1,2].**
>
> Thanks! We highly recommend the reviewer to see Appendix.F in the revision for more detailed comparison and discussion for InfoMax and other methods.
>
> Formulation. We would like to clarify that our major novelty lies in formulating the data pruning problem into a combinatorial optimization problem by jointly considering the intra-sample informativeness (importance) and inter-sample informativeness (redundancy). Moreover, we designed an efficient solver for this combinatorial problem. Finally, InfoMax achieves superior performance across various scenarios.
>
> While D2-Pruning [1] and Dos [2] also combine score-based and diversity-based methods, their approaches are quite different. Dos [2] is a scheme designed for OOD scenarios, with the idea of partitioning the feature space and selecting the most significant samples from each region. We have added the citation of [2] to the related work. However, the Dos selection paradigm cannot ensure that samples within each region are diverse enough. InfoMax can better balance the complex relationships among different factors while considering diversity and importance.
>
> D2-Pruning [1] uses a greedy selection method, picking the highest-scoring nodes first and lowering the scores of nearby nodes to manage redundancy. This greedy approach leads to less optimal results, while InfoMax can better optimize the overall information in the dataset. Our method uses a unified combinatorial optimization framework that effectively integrates sample information and diversity.
>
> [1] D2 pruning: Message passing for balancing diversity \& difficulty in data pruning. ICLR 2024.  [2] Dos: Diverse outlier sampling for out-of-distribution detection. ICLR 2024.
>
>
>
> **Weakness-3: The paper does not include comparisons with some relevant baseline methods, such as geometry-based methods.**
>
> Thank you for your suggestions to strengthen this work!
> We have added the results of the standard geometry-based method (K-center) in Table 1. Compared with existing hybrid methods, K-center's performance is somehow disadvantaged.
>
> **Question-1-1: Why InfoMax better than D2-Pruning?**
>
> This is an insightful question!
>
> D2-Pruning is a graph-inspired greedy selection method. In this framework, node values represent a sample's importance, while edges capture the similarities between samples. The data pruning process is formulated as a greedy iterative node selection procedure, where at each step, nodes with the highest scores are selected, and the scores of neighboring nodes are reduced to account for redundancy. However, due to its greedy selection process, the algorithm is prone to getting stuck in suboptimal solutions, making it challenging to maintain a proper balance between importance and diversity.
>
> In contrast, by formulating the maximizing the sample-wise informativeness while minimizing the redundancy, InfoMax forms a global optimization pipeline for data pruning from an information perspective. Therefore, while D2-pruning can often get trapped in local solutions, InfoMax aims to find the globally most informative subset. Moreover, InfoMax is equipped with an efficient proximal gradient-based solver with guaranteed convergence, leading to consistently superior results.
>
> To gain a deeper understanding, we conducted experiments with a selection ratio of 10\% on ImageNet as our research scenario. We performed a quantitative analysis of InfoMax and D2-Pruning based on the final coreset measurements, including mean redundancy and mean informativeness. Specifically, mean redundancy is defined as the average similarity among all samples, while mean informativeness is defined as the average sample-wise score value. The coreset found by InfoMax can have higher information and lower redundancy, leading to better performance for the model trained on the coreset.
>
> -|Mean-informativeness ($\uparrow$)|Mean-redundancy ($\downarrow$)|Top-1 Acc($\uparrow$)
> ---|---|---|---
> D2-Pruning|0.491|0.292|55.6
> InfoMax|0.563|0.216|59.0
>
>
> **Question-1.2: The reviewer also has concerns about the convex relaxation used in InfoMax.**
>
> Great question! Thank you!
>
> Convex relaxation is a common technique used in solving optimization problems. It expands the possible solutions and turns the problem into a continuous one, which makes it easier to find optimal solutions using gradient information. This helps avoid the pitfalls of greedy algorithms, which often get stuck in local optima. Our results show that InfoMax performs better in practice than older methods like D2-Pruning.

---

> > ### Comment · Reviewer_J8Fq · 2024-11-26
> > **Response to rebuttal**
> >
> > Thanks for the responses to my comments.
> >
> > It makes sense to me that the optimal solution to Eq. 7 is achievable. However, how can you ensure that the solution obtained by solving Eq. 7, followed by selecting the top-k, is equivalent to the optimal solution for Eq. 2?

---

> ### Author Response · Authors · 2024-11-28
> **Response to Reviewer J8Fq**
>
> Thank you for your reply and the discussion! We are happy to address your concerns.
>
> ---
>
> **Question: The guarantee of the error gap between the solution of the original discrete quadratic programming problem Eq.2 and that of the slack problem Eq.7.**
>
> One of the key contributions of InfoMax is its transformation of the coreset selection problem into the original discrete quadratic programming problem defined in Equation 2, viewed from an information-theoretic perspective. Given the high dimensionality of the problem in Equation 2, solving it presents significant challenges. To address this, InfoMax has introduced the efficient InfoMax-Solver, which effectively tackles the slack problem outlined in Equation 7. This innovation enables us to quickly obtain satisfactory solutions.
>
> The convex relaxation from Equation 2 to Equation 7 is one of the most common approaches in solving large-scale integer programming. At present, there is a gap between the optimal solution of the original quadratic integer programming problem and that of the slack problem. Analyzing the bounds of this gap is a particularly challenging topic, and previous work on this issue can be found in references [1, 2, 3]. In addition, we provide some empirical analysis of this gap.
>
> In the table below, we present some results obtained before submission. We compare the performance of directly using the integer programming solver CPLEX to solve Equation 2. Our experimental scenarios involve image classification tasks on CIFAR-10 and ImageNet-1K, with a coreset selection rate of 20%.
>
> On CIFAR-10, while directly solving Equation 2 provides a slight performance improvement, the overall time cost is prohibitively high, taking over 10,000 times longer than the InfoMax-Solver. In the larger-scale ImageNet-1K, the InfoMax-Solver achieved optimal performance in just 1.7 minutes, whereas using CPLEX to solve Equation 2 becomes unmanageable, with a time cost exceeding 7 days.
>
> This experiment highlights the necessity and efficiency of the InfoMax-Solver.
>
>
> Method|Dataset|Performance|Time-cost
> ---|---|---|---
> InfoMax-solver| CIFAR-10 (50000 data)| 92.7|11s
> CPLEX| CIFAR-10 (50000 data) | 92.9 | 36.2 hours
> InfoMax-solver| ImageNet-1K (1M data)| 66.5|1.7 min
> CPLEX| ImageNet-1K (1M data) | NAN | NAN
>
> ---
>
> [1]. Proximity in Concave Integer Quadratic Programming. Mathematical Programming
>
> [2]. Some proximity and sensitivity results in quadratic integer programming. Mathematical Programming
>
> [3]. The relationship between integer and real solutions of constrained convex programming. Mathematical Programming

---

> > ### Comment · Reviewer_J8Fq · 2024-11-29
> >
> > Thanks for the explanation, it has addressed my concerns. I have increased my rating to 8.

---

> > > ### Author Response · Authors · 2024-11-29
> > > **Sincere Gratitude to Reviewer J8Fq**
> > >
> > > We sincerely appreciate your response and are grateful for your recognition of our work! If you have any further concerns, please don't hesitate to let us know. We are more than happy to address your questions!
> > >
> > > We wish you all the best!
> > >
> > > Best regards,
> > >
> > > Submission1258 Authors

---

### Author Response · Authors · 2024-11-23
**General Response to ACs and Reviewers**

Dear Reviewers and ACs:

We sincerely appreciate your constructive comments and insightful reviews, which have significantly contributed to enhancing our work. We have thoroughly considered all your suggestions and made substantial revisions to our previous draft, with the main changes highlighted in blue.

Specifically, we have made the following changes:

    1. Fixed some typos.

    2. Rephrase Sec.4.4 in the main paper.

    3. Added comparison with a standard geometry-based method in the experiments in the main paper.

    4. Added some new symbolic conventions in the appendix.

    5. Added clarification and explanation for some symbols.

    6. Added ablations for hyper-parameters on classification tasks and LLM-SFT tasks in the appendix.

    7. Added error STD values in the appendix.

    8. Added detailed derivation of InfoMax solver in the appendix.

    9. Added the discussion about why InfoMax outperforms $D^2$-Pruning in the appendix.

Thank you very much again!

Best regards,
Authors of Paper1258

---

### Author Response · Authors · 2024-12-01
**Grateful Response to all Reviewers**

We would like to express our gratitude to all the reviewers for their hard work and dedication and for their contributions to the academic community.

As the discussion phase is coming to a close, if you have any questions or concerns regarding the article, we are more than happy to address them promptly!

Once again, thank you!

Best regards,

Submission1258 Authors

---

### Meta-Review · Area_Chair_PKhH · 2024-12-13

**Metareview:**

This paper focuses on the topic of data pruning. It presents InfoMax which is an innovative data pruning method aimed at maximizing the informational value of selected samples while reducing redundancy. This paper approaches the research problem by formulating it as a discrete quadratic programming task, and addresses it using an efficient gradient-based optimization technique. This paper is well-organized, with clear descriptions and justifications of complex concepts. Besides, the proposed method achieves promising performance on multiple benchmarks. The main weakness is reflected in its computational costs when the data scale is large. Overall, this is a good submission and makes solid contributions to the data pruning field. AC therefore recommends accepting it.

**Additional Comments On Reviewer Discussion:**

The reviewer emphasized that the paper exceeded the required number of pages, which was a violation. This somewhat affected the reviewer's judgment of the paper's quality. AC reviewed the paper and reported the issue to SAC. After consideration, the paper entered the review process normally. Therefore, the following additional comments are unrelated to the page limit issue.

There are six reviewers provide insightful comments on this work. The discussions and changes are summarized below.

- Reviewer J8Fq raised concerns about notation issues, lack of novelty in motivation, missing baseline comparisons, and equivalence of solutions. The rebuttal well addresses the concerns. The reviewer acknowledges that the current form is satisfactory.
- Reviewer nGro provided questions on empirical comparisons, data generalization, more in-depth analysis, and some minor comments. The authors provided detailed responses accordingly, which handled the questions properly.
- Reviewer HVmb mainly worried that this work lacks enough insights and novelty. The rebuttal provided a detailed comparison and analysis between this work and previous work (e.g., D2-pruning). The issue also was raised by other reviewers. AC checked the work and acknowledged the advancement of this work over previous works.
- Reviewer tXt5 commented about computational efficiency, hyperparameter sensitivity, and overfitting risk. The latter two were resolved in rebuttal. Computational costs are actually still high, which is a weakness of the proposed method as mentioned.
- Reviewer kwDs pointed out the issues of incomplete ablation study and analysis, which are addressed during rebuttal.
- Reviewer 2PGZ mainly questioned the idea novelty. The authors provide detailed explanations about the difference between this work and prior graph methods, from objectives and optimization.

Based on the above, AC considers that this paper overall makes solid contributions to data pruning (coreset selection), which meets the acceptance standards.

---

### Decision · Program_Chairs · 2025-01-22

Accept (Poster)